# A General Framework for Fair and Robust Regression

**Wenhai Cui** [* 1]   **Xiaoting Ji** [* 1]   **Wen Su** [2]   **Xingqiu Zhao** [1]

## Abstract

Fair regression methods typically rely on squared error loss, making them fragile under heavy tailed noise. We propose a general framework for robust regression under demographic parity (DP) that applies to a wide class of M-estimators, including Cauchy, Huber, least absolute deviation, quantile, and Tukey losses. We propose an optimal fair transformation that guarantees DP while achieving the minimum population risk among all rank preserving fair predictors. We also establish convergence rates for the resulting estimators. To balance fairness and predictive accuracy, we develop an interpolation scheme whose risk decreases while unfairness grows linearly with the interpolation parameter. The proposed framework can be further extended to conditional DP to account for legitimate covariates. Extensive simulation studies and real data applications show clear improvements over existing fair regression approaches in both robustness and predictive performance.

## 1. Introduction

### 1.1. Background and Motivation

Since machine learning algorithms are increasingly utilized in high-stakes domains, such as credit scoring and criminal justice, their robustness and fairness have received increased attention. Real-world data in these sectors frequently suffer from heavy-tailed errors and systemic biases. A representative example is the communities and crime (CRIME) dataset (Redmond & Baveja, 2002), which comprises socioeconomic features from 1,994 communities across the United States. The primary objective is to predict the crime rate

based on community-level information. However, empirical analysis suggests that standard least squares regression may be inadequate for this task.

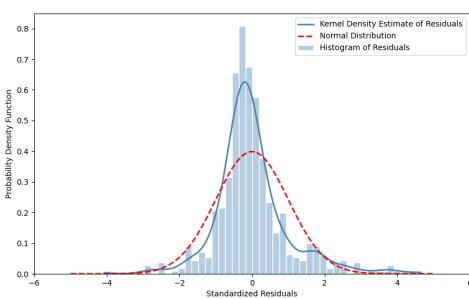

*(a)* Probability density function plot of standardized residuals.

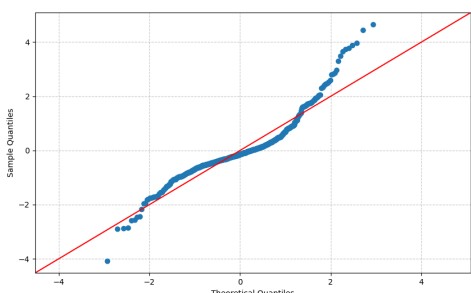

*(b)* Normal Q-Q plot of standardized residuals.

*Figure 1.* Diagnostic analysis of standardized residuals for the deep neural network estimator (trained on 70% of the CRIME dataset using squared error loss) with excess kurtosis of $\gamma_2 = 4.11$ and skewness of $\gamma_1 = 0.925$ (theoretical values are $\gamma_2 = 0$ and $\gamma_1 = 0$ for the standard normal distribution).

The residual diagnostics in Figure 1 reveal heavy-tailed residuals in the CRIME data. Specifically, the density plot of standardized residuals in Figure 1(a) exhibits a heavy-tailed distribution, characterized by a high excess kurtosis of $4.11$ and a positive skewness of $0.925$. The Q-Q plot in Figure 1(b) shows that the sample quantiles noticeably deviate from the theoretical reference line, particularly in the upper tail. Furthermore, the regression estimator yields an unfairness level of $0.3856$. Therefore, to simultaneously address the

---
[*]Equal contribution [1]Department of Applied Mathematics, The Hong Kong Polytechnic University [2]Department of Biostatistics, City University of Hong Kong. Correspondence to: Wen Su <w.su@cityu.edu.hk>, Xingqiu Zhao <xingqiu.zhao@polyu.edu.hk>.

*Proceedings of the 43rd International Conference on Machine Learning*, Seoul, South Korea. PMLR 306, 2026. Copyright 2026 by the author(s).

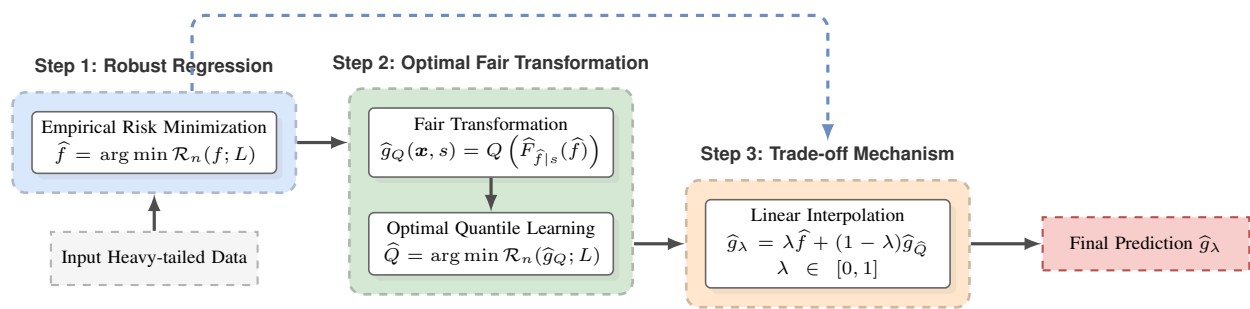

*Figure 2.* Overview of the proposed framework

heavy-tailed errors and unfairness, it is important to study robust regression under the demographic parity constraint.

Incorporating demographic parity constraints into robust regression poses several challenges. First, estimating the optimal fair predictor under general robust loss functions remains unexplored. Chzhen et al. (2020) derived a closed-form solution for the squared error loss by transforming group-specific predictors to their Wasserstein barycenter, and Liu et al. (2022) extended this method to quantile regression. However, there are no theoretical guarantees that such a solution minimizes a robust loss function under the demographic parity constraint. Second, decision-makers often require a flexible mechanism to balance fairness and accuracy, rather than being limited to two mutually exclusive options that permit nothing in between: an unconstrained risk minimizer or a perfectly fair predictor. Finally, the demographic parity constraint unconditionally eliminates dependence of predictor on sensitive attributes, ignoring the information from explanatory variables that are correlated with sensitive attributes. Therefore, a rigorous fairness framework must incorporate conditional demographic parity (CDP) to distinguish between spurious bias and valid feature dependence.

### 1.2. Our Contributions

We propose a post processing framework that transforms a robust predictor to satisfy demographic parity while preserving prediction accuracy. Our main contributions are summarized as follows.

**(i) General Robust Framework and Optimality:** We propose a general fair regression method that accommodates a broad class of robust loss functions (e.g., Cauchy, Huber, LAD, Quantile, and Tukey's biweight loss). We prove that our proposed transformation is the minimizer of the robust risk among all fair and rank-preserving predictors. To the best of our knowledge, this is the first work to establish such optimality guarantees for general robust loss functions.

**(ii) Trade-off Mechanism and Conditional Fairness:** We introduce an interpolation predictor whose risk decreases while unfairness grows linearly with the interpolation parameter. This guarantee allows policymakers to transparently balance predictive risk and fairness. Furthermore, we extend our approach to conditional demographic parity (CDP), allowing the model to account for valid explanatory variables while mitigating bias.

**(iii) Convergence Rates:** We establish a finite-sample convergence rate for the excess risk of the proposed estimator. Specifically, when the base predictor is parameterized by a deep neural network (DNN), we derive explicit rates that characterize the dependence on the input dimension and smoothness of true quantile function, ensuring the statistical consistency of our post-processing framework.

### 1.3. Related work

**Robust Regression.** The classic regression estimator under the squared loss is sensitive to heavy-tailed error distributions. To address this issue, robust loss functions have been extensively studied, including the Huber loss (Huber, 1973), Least Absolute Deviation (LAD) (Bassett Jr & Koenker, 1978), and Tukey's biweight loss (Beaton & Tukey, 1974). Recently, Fan et al. (2017) explored high-dimensional robust regression using the Huber loss, while Shen et al. (2021) established nonasymptotic error bounds for robust nonparametric regression estimators parameterized by using DNNs. Although these methods improved predictive reliability, they do not account for fairness constraints.

**Fair Regression.** Methodologies for mitigating bias in regression typically fall into three categories: pre-processing, in-processing, and post-processing. Specifically, pre-processing methods aim to remove bias from training data by learning fair representations (Zemel et al., 2013). In-processing methods incorporate fairness constraints into the training objective (Zafar et al., 2017). Post-processing methods transform the pre-trained predictor to satisfy fairness constraints. This approach is particularly attractive for "black-box" models where retraining is computationally infeasible. Recently, Chzhen et al. (2020) leveraged the optimal transport theory to transform group-specific predictors into their Wasserstein barycenter. This provides a closed-

form optimal solution for the squared loss. Liu et al. (2022) extended the method to quantile regression. However, current post-processing work is mostly limited to the squared loss. There is a lack of a unified framework that guarantees fairness for general robust loss functions.

## 2. Preliminaries

### 2.1. Robust Regression

Consider a nonparametric regression model:

$$Y = f_0(\boldsymbol{X}, S) + \eta, \tag{1}$$

where $Y \in \mathbb{R}$ is the response, $(\boldsymbol{X}, S)$ is a d-dimensional vector, $\boldsymbol{X} \in \mathcal{X}$ denotes covariates, $S \in \mathcal{S}$ represents a categorical sensitive attribute (e.g., race or gender), $f_0 : \mathcal{X} \times \mathcal{S} \to \mathbb{R}$ is the unknown regression function, and $\eta$ is an unobservable error term independent of $(\boldsymbol{X}, S)$, assumed to have a finite $p$-th moment. We aim to estimate the unknown target regression function $f_0$ given $(\boldsymbol{X}, S, Y)$.

To accommodate heavy-tailed errors, we consider the following robust loss function:

- Cauchy loss: $L(a, y) = \log\left\{1 + \kappa^2(a - y)^2\right\}, (a, y) \in \mathbb{R}^2$, for some $\kappa > 0$.

- Huber loss: For a tuning parameter $\zeta > 0$,

$$L(a, y) = \begin{cases} \frac{1}{2}(a - y)^2 & \text{if } |a - y| \leq \zeta, \\ \zeta|a - y| - \frac{1}{2}\zeta^2 & \text{otherwise.} \end{cases}$$

- Least absolute deviation (LAD) loss: $L(a, y) = |a - y|, (a, y) \in \mathbb{R}^2$.

- Quantile loss: For a quantile level $\tau \in (0, 1)$, $L(a, y) = \rho_\tau(a - y)$, where $\rho_\tau(u) = u(\tau - \mathbb{I}(u < 0))$.

- Tukey's biweight loss: For a truncation parameter $c > 0$,

$$L(a, y) = \begin{cases} \frac{c^2}{6}\left[1 - \left(1 - \left(\frac{a-y}{c}\right)^2\right)^3\right] & \text{if } |a - y| \leq c, \\ \frac{c^2}{6} & \text{otherwise.} \end{cases}$$

For any robust loss function $L$, the risk is defined as

$$\mathcal{R}(f; L) = \mathbb{E}_{\boldsymbol{Z}}[L(f(\boldsymbol{X}, S), Y)],$$

where $\boldsymbol{Z} = (\boldsymbol{X}, S, Y)$. Our objective is to estimate a measurable function $f^* : \mathcal{X} \times \mathcal{S} \to \mathbb{R}$ satisfying:

$$f^* = \arg\min_f \mathcal{R}(f; L) = \arg\min_f \mathbb{E}_{\boldsymbol{Z}}[L(f(\boldsymbol{X}, S), Y)]. \tag{2}$$

Under appropriate regularity conditions, the optimal solution $f^*$ corresponds to the true regression function $f_0$ in (1) (Shen et al., 2021).

### 2.2. Motivation

We begin with the definition of demographic parity, a widely used fairness criterion.

**Definition 2.1** (Demographic Parity). A predictor $f : \mathcal{X} \times \mathcal{S} \to \mathbb{R}$ satisfies demographic parity (DP) if and only if for every pair of sensitive attributes $s, s' \in \mathcal{S}$ and $t \in \mathbb{R}$,

$$P(f(\boldsymbol{X}, S) \leq t \mid S = s) = P(f(\boldsymbol{X}, S) \leq t \mid S = s').$$

Demographic parity requires that the distribution of $f(\boldsymbol{X}, S)$ does not depend on the sensitive attribute $S$. Our goal is to learn a predictor that minimizes a robust risk while strictly satisfying the DP constraint:

$$\min_{f \text{ satisfies DP}} \mathcal{R}(f; L). \tag{3}$$

## 3. Methodology

### 3.1. Demographic Parity Transformation

The following theorem characterizes a general class of predictors that satisfy DP.

**Theorem 3.1** (Closed form for demographic parity Transformation). *Assume the conditional cumulative distribution function (CDF) of $f^*(\boldsymbol{X}, S)$ given $S = s$, denoted by $F_{f^*|S=s}$, is continuous for all $s$. For any quantile function $Q$, the transformed predictor*

$$g_Q^*(\boldsymbol{x}, s) = Q\left(F_{f^*|s}(f^*(\boldsymbol{x}, s))\right)$$

*satisfies demographic parity.*

- **Generalization:** Theorem 3.1 establishes that the predictor via the transformation satisfies demographic parity for any quantile function $Q$. This result significantly generalizes Theorem 2.3 in Chzhen et al. (2020), which restricted the target distribution to the Wasserstein barycenter, i.e., $\bar{Q} = \sum_{s' \in \mathcal{S}} p_{s'} Q_{f^*|s'}$.

- **Connection with Optimal Transport:** From an optimal transport perspective, in the one-dimensional setting, the map $T(\cdot) = Q \circ F_{f^*|s}(\cdot)$ corresponds precisely to the optimal transport map that transports the conditional distributions of $f^*(\boldsymbol{X}, S)$ to the target distribution characterized by $Q$.

### 3.2. Toy Example: Suboptimality of $\bar{Q} \circ F_{f^*|s}(\cdot)$

We present a toy example to demonstrate that the solution based on the Wasserstein barycenter $\bar{Q}$ (Chzhen et al., 2020) is suboptimal under LAD loss.

Consider a regression setting, i.e.,

$$Y = X + \eta,$$

where $\eta \sim \mathcal{N}(0, \sigma^2)$. Let the sensitive attribute $S \in \{0, 1\}$ have marginal probabilities $P(S = 0) = 0.9$ and $P(S = 1) = 0.1$. The conditional distribution of $X$ given $S$ follows

$$X \mid S = 0 \sim \mathcal{U}(0, 1), \quad X \mid S = 1 \sim \mathcal{U}(2, 3).$$

The optimal unconstrained predictor is $f^*(x) = x$. The corresponding conditional CDFs are:

$$F_{f^*|S=0}(t) = t, t \in [0, 1]; \quad F_{f^*|S=1}(t) = t - 2, t \in [2, 3].$$

The corresponding conditional quantile functions are:

$$Q_{f^*|S=0}(u) = u, Q_{f^*|S=1}(u) = u + 2, u \in [0, 1].$$

The resulting predictor $h_{\bar{Q}}$ by $\bar{Q} \circ F_{f^*|s}$ is

$$h_{\bar{Q}} = \begin{cases} \sum_{s' \in \mathcal{S}} p_{s'} Q_{f^*|s'} \left( F_{f^*|S=0}(f^*(x)) \right), & \text{if } S = 0, \\ \sum_{s' \in \mathcal{S}} p_{s'} Q_{f^*|s'} \left( F_{f^*|S=1}(f^*(x)) \right), & \text{if } S = 1. \end{cases}$$
$$= \begin{cases} x + 0.2, & \text{if } S = 0, \\ x - 1.8, & \text{if } S = 1. \end{cases}$$

To simplify the illustration, we assume that $\sigma = 0.1$ and compute the LAD risk of $h_{\bar{Q}}$:

$$\mathcal{R}(h_{\bar{Q}}, L) = \mathbb{E}[|f^*(X) - h_{\bar{Q}}(X, S)|] \approx 0.362.$$

Consider a specific quantile function $Q(u) = u$. Thus, we get the fair predictor $h_Q$

$$h_Q(x, s) = \begin{cases} Q\left( F_{f^*|S=0}(f^*(x)) \right) = x, & \text{if } S = 0, \\ Q\left( F_{f^*|S=1}(f^*(x)) \right) = x - 2, & \text{if } S = 1. \end{cases}$$

We then compute the LAD risk for $h_Q$:

$$\mathcal{R}(h_Q, L) \approx 0.272.$$

This result ($\mathcal{R}(h_Q, L) \approx 0.272 < 0.362 \approx \mathcal{R}(h_{\bar{Q}}, L)$) implies that the transformation $\bar{Q} \circ F_{f^*|s}$ is suboptimal for LAD loss.

### 3.3. Optimal Quantile Learning

This toy example motivates us to estimate the optimal $Q$, rather than relying on the fixed $\bar{Q}$. We define the optimal quantile function as:

$$Q^* = \arg \min_{Q \in \mathcal{Q}} \mathcal{R}(g_Q^*; L), \tag{4}$$

where $\mathcal{Q}$ denotes a space of quantile functions and $g_Q^*$ is given in Theorem 3.1.

Since a quantile function $Q \in \mathcal{Q}$ is required to be monotonically increasing, we approximate the space using monotone I-splines. We divide the domain $[0, 1]$ into $m_n + 1$ sub-intervals using the knot sequence $0 = t_1 = \cdots = t_d <$

$t_{d+1} < \cdots < t_{m_n+d} < t_{m_n+d+1} = \cdots = t_{m_n+2d} = 1$, where $d$ denotes the order of the I-splines. Let $\{I_j(\tau)\}_{j=1}^{J_n}$ be the I-spline basis functions, where $J_n = m_n + d$. Define the functional space for the candidate quantile functions as:

$$\mathcal{Q}_n = \left\{ \alpha_0 + \sum_{j=1}^{J_n} \alpha_j I_j(\tau) : \alpha_j \geq 0 \text{ for } j = 1, \ldots, J_n \right\}.$$

where the unconstrained intercept $\alpha_0$ determines the location parameter, and the non-negative coefficients $\{\alpha_j\}_{j=1}^{J_n}$ ensure monotonicity. Let $\mathbf{I}(\tau) = (1, I_1(\tau), \ldots, I_{J_n}(\tau))^T$ and $\boldsymbol{\alpha} = (\alpha_0, \alpha_1, \ldots, \alpha_{J_n})^T$. The quantile function is thus parameterized as $Q(\tau) = \mathbf{I}(\tau)^T \boldsymbol{\alpha}$.

### 3.4. Estimation Procedure

Consider independent and identically distributed samples $\{(\boldsymbol{X}_i, S_i, Y_i)\}_{i=1}^n$ drawn from the joint distribution of $\{\boldsymbol{X}, S, Y\}$. We define the empirical risk of a predictor $f$ with respect to a loss function $L$ as:

$$\mathcal{R}_n(f; L) = \frac{1}{n} \sum_{i=1}^n L\left( f(\boldsymbol{X}_i, S_i), Y_i \right).$$

The proposed estimation procedure proceeds in three steps below.

**Step 1: Robust Regression.** First, we obtain a baseline predictor $\widehat{f}$ by minimizing the empirical risk over a pre-specified function class $\mathcal{F}$,

$$\widehat{f} = \arg \min_{f \in \mathcal{F}} \mathcal{R}_n(f; L).$$

**Step 2: Fairness Transformation.** Next, we construct a demographic parity-compliant predictor $\widehat{g}_Q$ via the transformation:

$$\widehat{g}_Q(\boldsymbol{x}, s) = Q\left( \widehat{F}_{\widehat{f}|s} \left( \widehat{f}(\boldsymbol{x}, s) \right) \right),$$

where $\widehat{F}_{\widehat{f}|s}$ denotes the empirical cumulative distribution function of the predictions $\widehat{f}(\boldsymbol{X}, S)$, given the sensitive attribute $S = s$.

**Step 3: Optimal Quantile Learning.** We estimate the optimal transformation $\widehat{Q}$ by minimizing the empirical risk over the spline space $\mathcal{Q}_n$:

$$\widehat{Q} = \arg \min_{Q \in \mathcal{Q}_n} \mathcal{R}_n(\widehat{g}_Q; L).$$

This optimization problem is solved with respect to the parameters $\boldsymbol{\alpha}$ subject to non-negativity constraints ($\alpha_j \geq 0, j \geq 1$) using the constrained L-BFGS-B algorithm (Byrd et al., 1995).

## 3.5. Trade-off Between Fairness and Risk Minimization

In practice, policymakers are often required to balance risk minimization and fairness. To rigorously quantify the unfairness of predictors, we adopt the Wasserstein-1 distance. Consider two distributions $\mu$ and $\nu$ on $\mathbb{R}$ with finite first moments. The Wasserstein-1 distance between them is defined as:

$$W_1(\mu, \nu) = \inf_{\gamma \in \Gamma(\mu, \nu)} \int_{\mathbb{R} \times \mathbb{R}} |x - y| \, d\gamma(x, y), \qquad (5)$$

where $\Gamma(\mu, \nu)$ denotes the set of all joint distributions (couplings) with marginals $\mu$ and $\nu$. A key property of the Wasserstein-1 distance in the one-dimensional case is that it admits a closed-form representation in terms of quantile functions: $W_1(\mu, \nu) = \int_0^1 |Q_\mu(\tau) - Q_\nu(\tau)| d\tau$, where $Q_\mu$ is the quantile function of distribution $\mu$.

Based on this metric, we formally define the unfairness measure associated with a predictor $f$.

**Definition 3.2** (Unfairness Measure). Let $\nu_s^f$ denote the conditional distribution of the predictor $f(\boldsymbol{X}, S)$ given $S = s$. The unfairness measure is defined as

$$\begin{aligned} \mathcal{U}(f) &= \sup_{s, s' \in \mathcal{S}} W_1(\nu_s^f, \nu_{s'}^f) \\ &= \sup_{s, s' \in \mathcal{S}} \int_0^1 \left| Q_{f|s}(\tau) - Q_{f|s'}(\tau) \right| d\tau, \end{aligned}$$

where $Q_{f|s}$ is the conditional quantile function of $f(\boldsymbol{X}, S)$ given $S = s$.

This measure quantifies the maximal conditional distributional discrepancy across different groups, serving as a rigorous metric for the degree of unfairness.

**Proposition 3.3.** *The unfairness measure vanishes, i.e., $\mathcal{U}(f) = 0$, if and only if for all $s, s' \in \mathcal{S}$, the Kolmogorov-Smirnov distance is zero, i.e.,*

$$\sup_{t \in \mathbb{R}} |P(f \leq t \mid S = s) - P(f \leq t \mid S = s')| = 0.$$

Consequently, the condition $\mathcal{U}(f) = 0$ implies that the predictor $f$ strictly satisfies demographic parity. Let $f^*$ denote the unconstrained risk minimizer, and let $\mathcal{U}(f^*)$ represent the baseline unfairness level associated with the optimal predictor $f^*$. We quantify the relative reduction in unfairness compared to the baseline $\mathcal{U}(f^*)$.

**Definition 3.4** ($\lambda$-Relative Fairness). A predictor $f$ is said to satisfy the $\lambda$-relative fairness if

$$\frac{\mathcal{U}(f)}{\mathcal{U}(f^*)} \leq \lambda, \text{ for } \lambda \in [0, 1]. \qquad (6)$$

The parameter $\lambda$ controls the unfairness level, requiring the predictor $f$ to retain at most a fraction $\lambda$ of the maximal

unfairness exhibited by the unconstrained risk minimizer $f^*$.

To achieve varying levels of relative fairness, we define the interpolated predictor as

$$g_\lambda^*(\boldsymbol{x}, s) = \lambda f^*(\boldsymbol{x}, s) + (1 - \lambda) g_{Q^*}^*(\boldsymbol{x}, s),$$

where the interpolation parameter $\lambda \in [0, 1]$. The predictor $g_\lambda^*$ exhibits the following properties:

(i) The predictor $g_\lambda^*$ smoothly interpolates between the perfectly fair predictor $g_{Q^*}^*$ (at $\lambda = 0$) and the unconstrained risk minimizer $f^*$ (at $\lambda = 1$).

(ii) Since $f^*$ and $g_{Q^*}^*$ are comonotonic, their convex combination $g_\lambda^*$ remains a strictly increasing transformation of $f^*$. Consequently, the intra-group ranking of individuals is invariant with respect to $\lambda \in [0, 1]$.

The following theorem establishes that the interpolated predictor $g_\lambda^*$ achieves $\lambda$-relative fairness.

**Theorem 3.5.** *Let $\mathcal{U}(f^*)$ denote the baseline unfairness of the unconstrained predictor $f^*$. For any trade-off parameter $\lambda \in [0, 1]$, the unfairness measure of the interpolated predictor $g_\lambda^*$ satisfies*

$$\frac{\mathcal{U}(g_\lambda^*)}{\mathcal{U}(f^*)} = \lambda.$$

Theorem 3.5 implies that the predictor $g_\lambda^*$ achieves $\lambda$-relative fairness, where $\lambda$ can be set by policy makers to satisfy specific fairness requirements.

**Theorem 3.6** (Monotonicity of Risk via Interpolation). *Assume the loss function $L(a, y)$ is convex with respect to the prediction $a$ for any fixed $y$ (e.g., the Huber, LAD, and quantile losses). Then, the risk function $J(\lambda) = \mathcal{R}(g_\lambda^*; L)$ is non-increasing over the interval $[0, 1]$.*

Theorems 3.5 and 3.6 imply that as the interpolation parameter $\lambda$ increases, we observe a reduction in risk accompanied by a strictly linear increase in unfairness.

## 4. Theoretical Results

### 4.1. Optimality of the Rank-Preserving Fair Predictor

In this section, we establish that the proposed transformation yields the optimal solution that minimizes the risk among all rank-preserving fair predictors. We first formally define rank-preserving predictors.

**Definition 4.1** (Rank-Preserving Predictors). A predictor $g$ is said to satisfy rank-preservation if, for any group $s \in \mathcal{S}$ and any pair of individuals $\boldsymbol{x}_i, \boldsymbol{x}_j \in \mathcal{X}$, the condition $f^*(\boldsymbol{x}_i, s) \geq f^*(\boldsymbol{x}_j, s)$ implies $g(\boldsymbol{x}_i, s) \geq g(\boldsymbol{x}_j, s)$.

This definition of rank-preservation implies that for each group $s$, $g(\cdot, s)$ must be a non-decreasing transformation of $f^*(\cdot, s)$, which ensures that the most qualified individuals (according to $f^*$) remain the most likely to be selected within their demographic group.

Let $\mathcal{G}_{\text{rank}}$ denote the class of all predictors $g : \mathcal{X} \times \mathcal{S} \to \mathbb{R}$ that simultaneously satisfy demographic parity and rank-preservation.

**Theorem 4.2** (Risk Minimization). *Assume that the conditions in Theorem 3.1 hold. Then, $g_{Q^*}^*$ achieves the minimum risk within the class $\mathcal{G}_{\text{rank}}$:*

$$\mathcal{R}(g_{Q^*}^*; L) = \inf_{g \in \mathcal{G}_{\text{rank}}} \mathcal{R}(g; L).$$

Theorem 4.2 implies that solving the risk minimization problem over the quantile space $\mathcal{Q}$ yields the optimal predictor within the function space $\mathcal{G}_{\text{rank}}$.

### 4.2. Risk Bounds for the Interpolated Predictor

The following theorem characterizes the performance of interpolation predictor $g_\lambda^*$ under robust loss functions.

**Theorem 4.3** (Risk Bounds). *Suppose that the loss function $L(\cdot, y)$ is Lipschitz continuous with constant $c_L > 0$. Then, the risk of the interpolated predictor $g_\lambda^*$ satisfies:*

$$\mathcal{R}(g_\lambda^*; L) \leq \mathcal{R}(f^*; L)$$
$$+ c_L(1 - \lambda)\mathbb{E}\left[|f^*(\boldsymbol{X}, S) - g_{Q^*}^*(\boldsymbol{X}, S)|\right].$$

Theorem 4.3 implies that the risk is bounded by the minimum risk and the difference between the fair and unconstrained predictors, which scales down linearly as $\lambda$ increases.

### 4.3. Convergence Analysis for Excess Risk

In this section, we establish the convergence rate for the proposed estimator $\widehat{g}_{\widehat{Q}}$. We require the following assumptions.

**Assumption 4.4.** (i) The true optimal quantile function $Q^*$ has bounded $r$-th derivative, where $r \geq 1$.

(ii) The number of subintervals in $[0, 1]$ satisfies $m_n = O(n^\nu)$ for $0 < \nu < 1/2$. Furthermore, we have $\max_{d+1 \leq i \leq m_n + d + 1} |t_i - t_{i-1}| = O(n^{-v})$ and

$$\frac{\max_{d+1 \leq i \leq m_n + d + 1} |t_i - t_{i-1}|}{\min_{d+1 \leq i \leq m_n + d + 1} |t_i - t_{i-1}|} \leq c_1,$$

uniformly for $n$ with a constant $c_1 > 0$.

(iii) $|L(u_1, y) - L(u_2, y)| \leq c_L|u_1 - u_2|$ for a constant $c_L > 0$.

(iv) There exist constants $c_3, \epsilon_{L,Q^*} > 0$ such that

$$\mathcal{R}(g_Q^*; L) - \mathcal{R}(g_{Q^*}^*; L) \leq c_3\|Q - Q^*\|_2^2, \quad (7)$$

for any $Q$ satisfying $\|Q - Q^*\|_2^2 \leq \epsilon_{L,Q^*}$, where $\|g\|_2^2 = \mathbb{E}_Z[g(Z)^2]$.

(v) The estimator $\widehat{f}$ satisfies $\|\widehat{f} - f^*\|_{L_2} = O_p(n^{-\mu})$, where $\mu \leq 1/2$.

(vi) For any $s \in \mathcal{S}$, the conditional density function of $f^*(\boldsymbol{X}, S)$ given $S = s$ is uniformly bounded.

(vii) For any $s \in \mathcal{S}$, it holds that $P(S = s) > 0$.

Assumptions (i) and (ii) are standard regularity conditions for spline approximation (Lu et al., 2007). Assumption (iii) is naturally satisfied by aforementioned robust loss functions. Assumption (iv) is a standard regularity condition in robust regression (Shen et al., 2021). It holds for smooth losses like Huber loss. For quantile or LAD losses, it is satisfied if noise has a bounded density function. Assumption (v) is a general condition. Assumptions (vi) and (vii) impose mild regularity conditions on the data distribution.

**Theorem 4.5** (Convergence Rate). *Under Assumption 4.4, for predictor $\widehat{g}_{\widehat{Q}}$, we have*

$$\mathcal{R}(\widehat{g}_{\widehat{Q}}, L) - \mathcal{R}(g_{Q^*}^*, L) = O_p\left(n^{-2\nu r} + n^{-\mu + \nu}\right),$$

*where $r$ determines the smoothness of the true function $Q^*$ and $0 < \nu < 1/2$.*

Consider the specific case where the base predictor $\widehat{f}$ is parameterized by DNNs. As shown by Shen et al. (2021), the estimation error satisfies $\|\widehat{f} - f^*\|_{L_2} = O_p(n^{-\mu_0})$, with

$$\mu_0 = \left(1 - \frac{1}{p}\right)\frac{\alpha}{d + \alpha},$$

where $d$ is the input dimension, $\alpha$ denotes the smoothness parameter of $f^*$, and the response $Y$ has a finite $p$-th moment. By selecting the smoothing parameter $\nu = \frac{\mu_0}{2r+1}$, the excess risk of the proposed fair predictor converges at the rate:

$$\mathcal{R}(\widehat{g}_{\widehat{Q}}, L) - \mathcal{R}(g_{Q^*}^*, L) = O_p\left(n^{-\left(1 - \frac{1}{p}\right)\frac{2r\alpha}{(d+\alpha)(2r+1)}}\right).$$

## 5. Extension to Conditional Demographic Parity

Demographic parity requires strict independence between predictions and sensitive attributes. However, there exist valid explanatory variables $W \in \mathcal{W}$ (e.g., education level) that are correlated with sensitive attributes, yet play a crucial role in the decision-making process. In such scenarios, enforcing demographic parity disregards the valid information contained in $W$. Therefore, we adopt the conditional demographic parity constraint.

**Definition 5.1** (Conditional Demographic Parity). A predictor $f$ is said to satisfy conditional demographic parity with respect to a discrete control variable $W \in \mathcal{W}$ if and only if, for all sensitive groups $s, s' \in \mathcal{S}$, every stratum $w \in \mathcal{W}$, and any threshold $t \in \mathbb{R}$:

$$P(f(\boldsymbol{X}, S, W) \leq t \mid S = s, W = w)$$
$$= P(f(\boldsymbol{X}, S, W) \leq t \mid S = s', W = w).$$

The conditional demographic parity constraint requires the predictor $f(\boldsymbol{X}, S, W)$ to be conditionally independent of the sensitive attribute $S$, given explanatory variables $W$.

To achieve CDP within our rank-preserving framework, we propose the stratified demographic parity transformation. Let $\mathcal{W} = \{w_1, \ldots, w_K\}$ be the finite set of control strata. We generalize the transformation by learning a stratum-specific optimal quantile function $Q_w$ for $w \in \mathcal{W}$. Then, the stratified predictor is defined as

$$g_{\mathcal{Q}_w}^{\mathrm{CDP}}(\boldsymbol{x}, s, w) = Q_w\left(F_{f^*|s,w}\left(f^*(\boldsymbol{x}, s, w)\right)\right), \quad (8)$$

where $F_{f^*|s,w}$ denotes the conditional cumulative distribution function of $f^*(\boldsymbol{X}, S, W)$ given $S = s$ and $W = w$. The risk can be expressed as:

$$\mathbb{E}\left[L(g_{Q_W}^{\mathrm{CDP}}(\boldsymbol{X}, S, W), Y)\right]$$
$$= \mathbb{E}\left[\sum_{w \in \mathcal{W}} \mathbb{I}\{W = w\} L(g_{Q_w}^{\mathrm{CDP}}(\boldsymbol{X}, S, W), Y)\right].$$

This formulation allows the global optimization problem to decompose into $K$ independent sub-problems. Thus, for each stratum $w$, we solve for the optimal quantile function $Q_w^*$ by minimizing the stratum-specific risk

$$Q_w^* = \arg\min_{Q_w \in \mathcal{Q}} \mathbb{E}\left[L(g_{Q_w}^{\mathrm{CDP}}(\boldsymbol{X}, S, W), Y) \cdot \mathbb{I}(W = w)\right].$$

Finally, the optimal fair predictor is constructed by assembling these stratum-specific solutions, i.e., $g_{Q_w^*}^{\mathrm{CDP}}(\boldsymbol{x}, s, w)$.

## 6. Numerical Studies

We compared our approach to two methods: the robust regression with deep neural networks (RDNN) (Shen et al., 2021), which focuses solely on robust regression without considering fairness constraint, and the fair regression with Wasserstein barycenters (FRWB) (Chzhen et al., 2020), a widely adopted benchmark that derives a closed-form fair predictor for the least squares loss. To ensure a fair comparison, we adapted the FRWB method by using the robust regression (Step 1 of Section 3.4) to obtain the initial estimator.

### 6.1. Simulations

To evaluate the finite-sample performance, we conducted a simulation study in a regression setting. The covariate

*Table 1.* Sample means and standard deviation (in parentheses) of empirical risk and unfairness for simulation across Cauchy loss with $\kappa = 1$, Huber loss with $\zeta = 1.345$, LAD loss, Quantile loss with $\tau = 0.75$ and Tukey loss with $c = 4.685$ over 200 repetitions.

| Loss | Method | Risk | Unfairness |
|---|---|---|---|
| Cauchy | RDNN | 0.7788 (0.0324) | 2.4669 (0.1962) |
| | FRWB | 1.2354 (0.0646) | 0.0125 (0.0015) |
| | **Ours** | **1.0469** (0.0443) | **0.0048** (0.0009) |
| Huber | RDNN | 0.9743 (0.0532) | 2.4809 (0.1612) |
| | FRWB | 1.5005 (0.0864) | 0.0127 (0.0015) |
| | **Ours** | **1.4214** (0.0772) | **0.0049** (0.0010) |
| LAD | RDNN | 1.1535 (0.0431) | 2.4669 (0.1766) |
| | FRWB | 1.7207 (0.0774) | 0.0126 (0.0014) |
| | **Ours** | **1.5718** (0.0681) | **0.0044** (0.0009) |
| Quantile | RDNN | 0.4561 (0.0180) | 3.7679 (0.1809) |
| | FRWB | 0.8029 (0.0290) | 0.0130 (0.0015) |
| | **Ours** | **0.7950** (0.0288) | **0.0045** (0.0011) |
| Tukey | RDNN | 0.7976 (0.0385) | 2.4701 (0.1920) |
| | FRWB | 1.2320 (0.0749) | 0.0126 (0.0015) |
| | **Ours** | **1.0909** (0.0512) | **0.0048** (0.0010) |

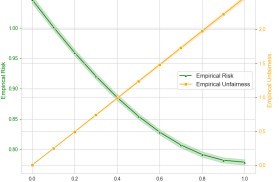 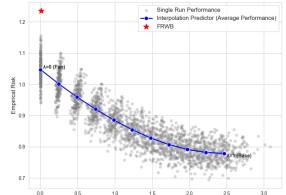

*(a)* Empirical risk and unfairness with interpolation hyperparameter $\lambda$.

*(b)* Illustrates the trade-off curve (blue line) about empirical risk and unfairness.

*Figure 3.* Performance evaluation of simulation over 200 independent trials under Cauchy loss with $\kappa = 1$.

vector $\boldsymbol{X} = (X_1, \ldots, X_5)^\top$ was generated from a standard multivariate normal distribution, i.e., $\boldsymbol{X} \sim \mathcal{N}(\boldsymbol{0}, \mathbf{I}_5)$. Conditional on $\boldsymbol{X} = \boldsymbol{x}$, the binary sensitive attribute $S \in \{0, 1\}$ was drawn from a Bernoulli distribution:

$$S \mid \boldsymbol{X} = \boldsymbol{x} \sim \mathrm{Bernoulli}(\pi(\boldsymbol{x})),$$

where $\pi(\boldsymbol{x}) = \frac{1}{1+\exp(-\boldsymbol{x}^\top \boldsymbol{\gamma})}$. We set the coefficient vector to $\boldsymbol{\gamma} = (1.0, -1.0, 0.5, 0.0, 0.0)^\top$. The response variable $Y$ was generated from the following linear model:

$$Y = \boldsymbol{X}^\top \boldsymbol{\beta}_0 + S + (2S + 0.5)\epsilon,$$

where the regression coefficients are $\boldsymbol{\beta}_0 = (1.5, -0.5, 0.2, 0.0, 0.0)^\top$, and the error term $\epsilon$ follows a standard normal distribution. Both the training and testing datasets consisted of 1,000 samples.

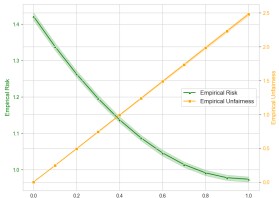
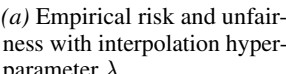

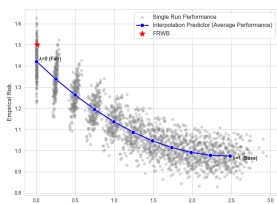

(a) Empirical risk and unfair-ness with interpolation hyper-parameter $\lambda$.

(b) Illustrates the trade-off curve (blue line) about empir-ical risk and unfairness.

*Figure 4.* Performance evaluation of simulation over 200 independent trials under Huber loss with $\zeta = 1.345$.

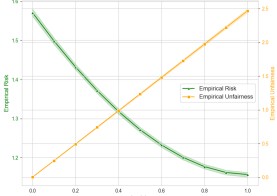
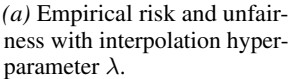

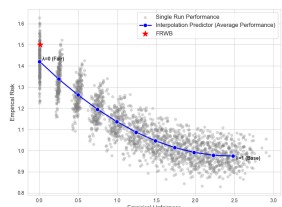

(a) Empirical risk and unfair-ness with interpolation hyper-parameter $\lambda$.

(b) Illustrates the trade-off curve (blue line) about empir-ical risk and unfairness.

*Figure 5.* Performance evaluation of simulation over 200 independent trials under LAD loss.

Table 1 shows that our method achieves the lowest level of unfairness compared to both the RDNN and FRWB. Furthermore, across various robust loss functions, our method consistently yields lower empirical risk than FRWB.

As illustrated in Figure 3(a), increasing the interpolation parameter $\lambda$ leads to a reduction in empirical risk with a linear increase in unfairness level, which is consistent with Theorem 3.5 and 3.6. In Figure 3(b), we observe that at the same level of unfairness, our method (blue line) achieves a significantly lower risk than the FRWB method (red star). This indicates that our approach provides a superior fairness-accuracy trade-off. Similar trends are observed in Figures 4 and 5 for the Huber and LAD losses, respectively. Additional simulation results are provided in Appendix A.

### 6.2. Real Data Application

We consider the communities and crime (CRIME) dataset (Redmond & Baveja, 2002). This dataset comprises socioeconomic data from 1,994 communities in the United States. The objective is to predict the crime rate based on community-level information. Following (Calders et al., 2013; Liu et al., 2022), we define the sensitive attribute $S$ as the majority racial group within each community.

Table 2 shows that our method obtains the lowest level of unfairness, demonstrating its capability to strictly enforce fairness constraints. Such significantly low unfairness im-

*Table 2.* Sample means and standard deviation (in parentheses) of empirical risk and unfairness for CRIME dataset across Cauchy loss with $\kappa = 1$, Huber loss with $\zeta = 1.345$, LAD loss, Quantile loss with $\tau = 0.75$ and Tukey loss with $c = 4.685$ over 30 repetitions.

| Loss | Method | Risk | Unfairness |
|---|---|---|---|
| Cauchy | RDNN | 0.0180 (0.0012) | 0.3993 (0.0309) |
| | FRWB | 0.0339 (0.0016) | 0.0647 (0.0190) |
| | **Ours** | 0.0352 (0.0024) | **0.0106** (0.0005) |
| Huber | RDNN | 0.0096 (0.0007) | 0.3950 (0.0309) |
| | FRWB | 0.0186 (0.0009) | 0.0646 (0.0175) |
| | **Ours** | 0.0195 (0.0014) | **0.0106** (0.0005) |
| LAD | RDNN | 0.0905 (0.0028) | 0.4000 (0.0282) |
| | FRWB | 0.1330 (0.0035) | 0.0615 (0.0172) |
| | **Ours** | **0.1294** (0.0049) | **0.0093** (0.0005) |
| Quantile | RDNN | 0.0418 (0.0019) | 0.4649 (0.0233) |
| | FRWB | 0.0662 (0.0024) | 0.0721 (0.0231) |
| | **Ours** | 0.0687 (0.0032) | **0.0107** (0.0007) |
| Tukey | RDNN | 0.0095 (0.0007) | 0.3967 (0.0312) |
| | FRWB | 0.0184 (0.0010) | 0.0641 (0.0163) |
| | **Ours** | 0.0192 (0.0013) | **0.0104** (0.0007) |

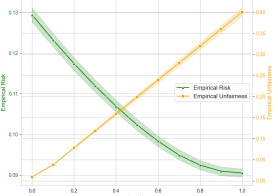
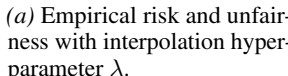

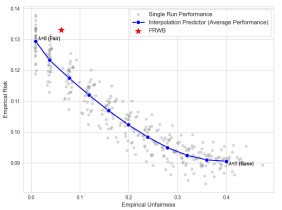

(a) Empirical risk and unfair-ness with interpolation hyper-parameter $\lambda$.

(b) Illustrates the trade-off curve (blue line) about empir-ical risk and unfairness.

*Figure 6.* Performance evaluation of CRIME dataset over 30 repetitions under LAD loss.

plies that our method imposes a more restrictive constraint on the solution space. In terms of empirical risk, our method remains highly competitive with FRWB. This is justifiable considering the substantially stricter fairness constraints satisfied by our method. At the same level of unfairness, Figure 6(b) shows that our method achieves a lower empirical risk compared to FRWB, indicating a superior trade-off. Furthermore, Figure 6(a) illustrates the trade-off dynamics: as $\lambda$ increases, the degree of unfairness grows linearly while the risk decreases, consistent with Theorems 3.5 and 3.6.

### 6.3. Adaptive Selection of the Trade-off Parameter

Assume there exists a budget function $\mathcal{B}(g_\lambda^*)$ determined by policymakers. For instance, let $\mathcal{B}(g_\lambda^*) = \mathcal{R}(g_\lambda^*; L)$. Our objective is to find an optimal trade-off for the pair

$[\mathcal{U}(g_\lambda^*), \mathcal{B}(g_\lambda^*)]$. We can identify the optimal $\lambda$ by the concept of knee point (Satopaa et al., 2011), which comes from multi-objective optimization. Let $u = \mathcal{U}(g_\lambda^*)$. Then, we can get $\lambda(u) = \frac{u}{\mathcal{U}(f^*)}$. This yields the utility-budget pair $[u, \mathcal{B}(g_{\lambda(u)}^*)]$. To simplify the notation, let $\mathcal{T}(u) = \mathcal{B}(g_{\lambda(u)}^*)$. The function $\mathcal{T}(u)$ represents the budget cost at unfairness level $u$. The knee point of $\mathcal{T}(u)$ maximizes its curvature, i.e,

$$u_0 = \arg\max \frac{\mathcal{T}''(u)}{[1 + \mathcal{T}'(u)^2]^{1.5}}.$$

Then, we get $\lambda(u_0) = \frac{u_0}{\mathcal{U}(f^*)}$. Mathematically, the knee point represents the farthest point of $\mathcal{T}(u)$ away from a straight line $y = 1 - x$. It indicates that as $\lambda$ decreases in $[0, \lambda(u_0)]$, although the unfairness level $\mathcal{U}(g_\lambda^*)$ decreases, the budget $\mathcal{B}(g_\lambda^*)$ increases at a faster rate than the reduction in $\mathcal{U}(g_\lambda^*)$. Thus, it is not worthy to achieve a marginal gain in fairness while paying a disproportionately high cost for $\lambda \in [0, \lambda(u_0)]$. We applied the method to a real-world CRIME dataset and obtained a proposed trade-off parameter of $\lambda(u_0) = 0.52$. The resulting trade-off lines are illustrated in Figure 7.

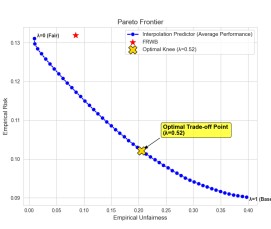

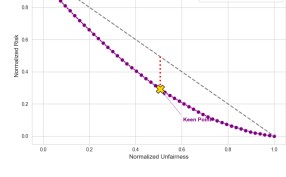

*(a) Pareto Frontier*

*(b) Normalized Pareto Frontier*

*Figure 7.* Empirical risk (LAD) and unfairness across various interpolation parameters $\lambda$ under real CRIME data. **(a)** The empirical trade-off between predictive risk and unfairness. The yellow marker indicates the optimal knee point ($\lambda = 0.52$) identified by the Kneedle algorithm, representing the most cost-effective operating point for real-world deployment. **(b)** The underlying mathematical mechanism for the optimal $\lambda$ selection. By mapping the frontier into a $[0, 1]$ normalized space, the Kneedle algorithm rigorously identifies the equilibrium point that maximizes the vertical distance to the reference chord.

## Acknowledgements

We thank the anonymous reviewers of ICML 2026 for their valuable suggestions. This research was supported in part by the Research Grant Council of Hong Kong (21215325), the National Natural Science Foundation of China (12401366, 12271459), and the research grants from City University of Hong Kong (9678373, 7020159).

## Impact Statement

This paper presents work whose goal is to advance the field of Machine Learning. There are many potential societal consequences of our work, none which we feel must be specifically highlighted here.

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

# A. Additional Numerical Studies

## A.1. Additional Simulation Results

This section presents additional simulation results across various loss functions. Notably, our method consistently achieves a lower empirical risk than FRWB. Furthermore, as the interpolation parameter $\lambda$ increases, we observe a clear trade-off where the empirical risk decreases while the unfairness measure increases.

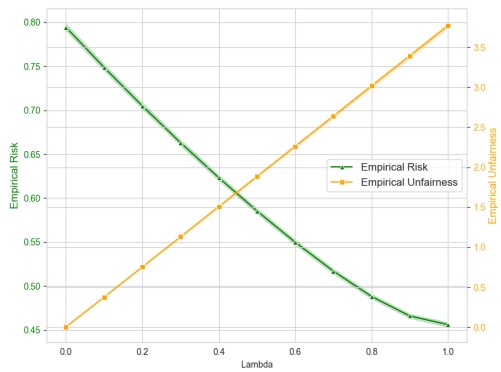
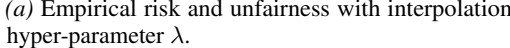

*(a)* Empirical risk and unfairness with interpolation hyper-parameter $\lambda$.

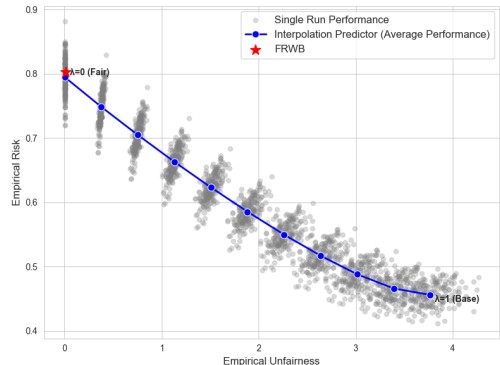

*(b)* Illustrates the trade-off curve (blue line) about empirical risk and unfairness.

*Figure 8.* Performance evaluation of simulation over 200 independent trials under quantile loss with $\tau = 0.75$.

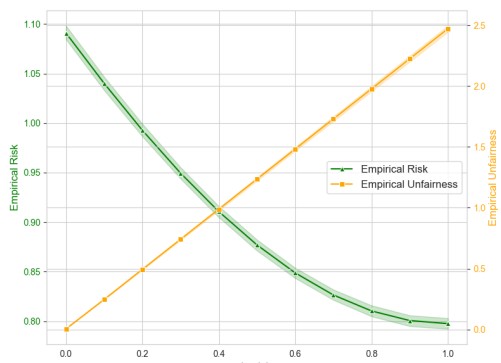

*(a)* Empirical risk and unfairness with interpolation hyper-parameter $\lambda$.

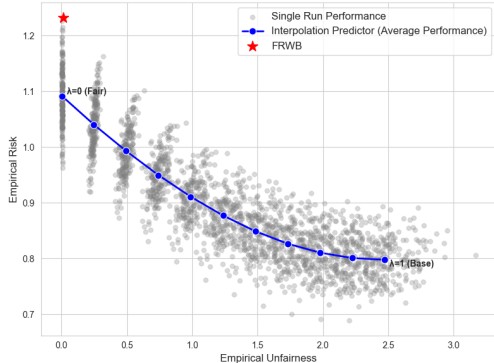

*(b)* Illustrates the trade-off curve (blue line) about empirical risk and unfairness.

*Figure 9.* Performance evaluation of simulation over 200 independent trials under Tukey loss with $c = 4.685$.

## A.2. Additional Real Data Results

This section presents supplementary experimental results on real-world data, demonstrating the efficacy and robustness of our proposed estimator under diverse loss functions. Figures 9(b), 10(b), 11(b), and 12(b) reveal that, for any fixed level of unfairness, our method outperforms FRWB by achieving significantly lower empirical risk. Additionally, Figures 9(a), 10(a), 11(a), and 12(a) confirm the theoretical predictions regarding the interpolation parameter $\lambda$: we observe a strict reduction in risk accompanied by a linear increase in unfairness.

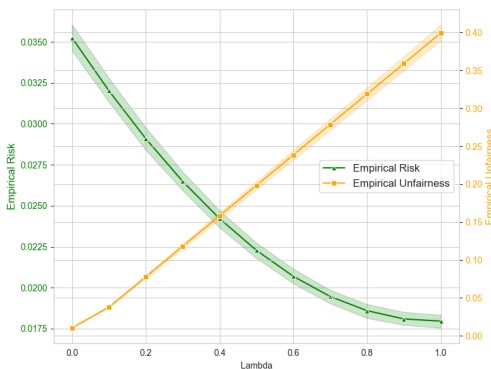
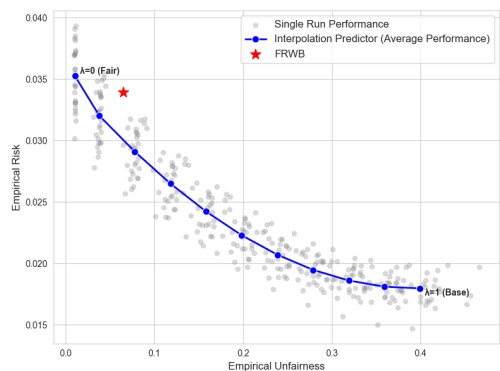

*(a)* Empirical risk and unfairness with interpolation hyper-parameter $\lambda$.

*(b)* Illustrates the trade-off curve (blue line) about empirical risk and unfairness.

*Figure 10.* Performance evaluation of CRIME dataset over 30 repetitions under Cauchy loss with $\kappa = 1$.

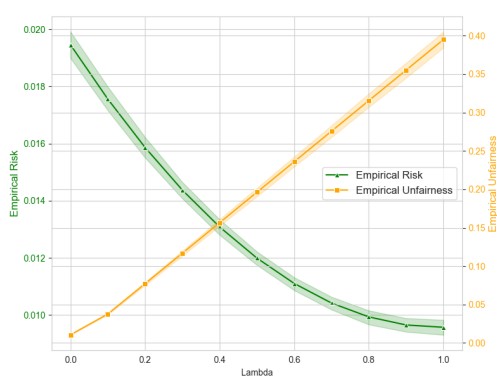
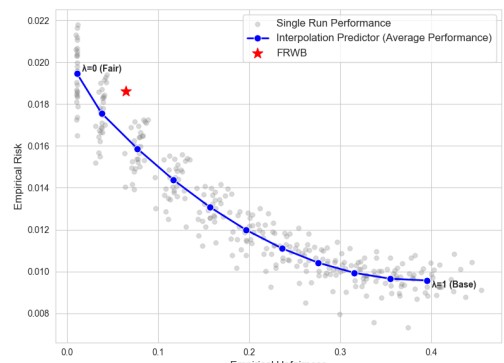

*(a)* Empirical risk and unfairness with interpolation hyper-parameter $\lambda$.

*(b)* Illustrates the trade-off curve (blue line) about empirical risk and unfairness.

*Figure 11.* Performance evaluation of CRIME dataset over 30 repetitions under Huber loss with $\zeta = 1.345$.

## B. Proof of Theorem 3.1

*Proof.* For any $s \in \mathcal{S}$, we have

$$
\begin{aligned}
&P(g_Q^*(\boldsymbol{X}, S) \le t \mid S = s) \\
=&P\left(Q \circ F_{f^*|S=s}\left(f^*(\boldsymbol{X}, S)\right) \le t \mid S = s\right) \\
=&P\left(F_{f^*|S=s}\left(f^*(\boldsymbol{X}, s)\right) \le Q^{-1}(t) \mid S = s\right).
\end{aligned}
$$

Since $F_{f^*|S=s}$ is the conditional CDF of the random variable $f^*(\boldsymbol{X}, S)$ given $S = s$, the random variable $U = F_{f^*|S=s}\left(f^*(\boldsymbol{X}, s)\right)$ follows a uniform distribution on $[0, 1]$, i.e., $U \sim \text{Uniform}(0, 1)$.

Therefore, we obtain:

$$
P\left(F_{f^*|S=s}\left(f^*(\boldsymbol{X}, s)\right) \le Q^{-1}(t) \mid S = s\right) = Q^{-1}(t).
$$

Since the result $Q^{-1}(t)$ depends only on $t$, but not on the sensitive attribute $s$, it holds that for any $s' \in \mathcal{S}$:

$$
\begin{aligned}
&P(g_Q^*(\boldsymbol{X}, S) \le t \mid S = s) \\
=&P(g_Q^*(\boldsymbol{X}, S) \le t \mid S = s') = Q^{-1}(t).
\end{aligned}
$$

Thus, $g_Q^*$ satisfies the definition of demographic parity. $\qquad\square$

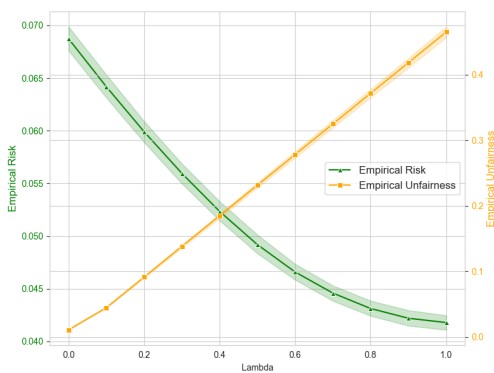

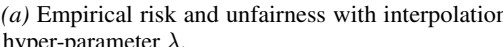

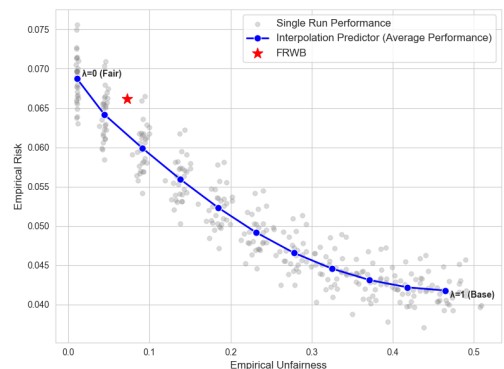

*(a)* Empirical risk and unfairness with interpolation hyper-parameter $\lambda$.

*(b)* Illustrates the trade-off curve (blue line) about empirical risk and unfairness.

*Figure 12.* Performance evaluation of CRIME dataset over 30 repetitions under Quantile loss with $\tau = 0.75$.

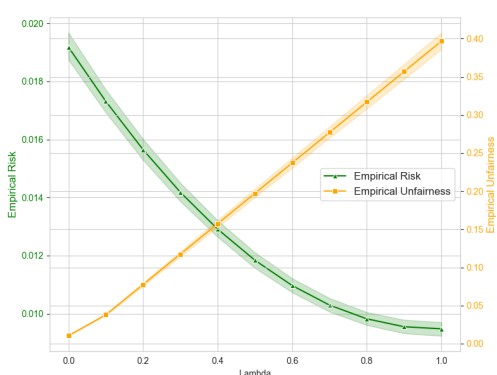

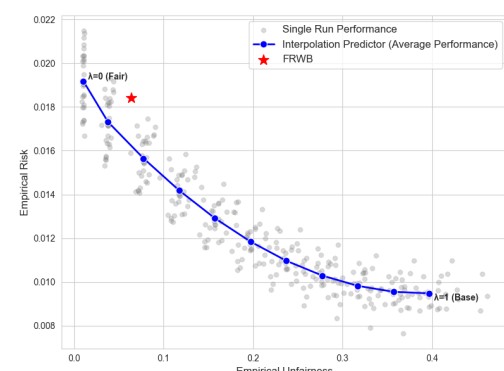

*(a)* Empirical risk and unfairness with interpolation hyper-parameter $\lambda$.

*(b)* Illustrates the trade-off curve (blue line) about empirical risk and unfairness.

*Figure 13.* Performance evaluation of CRIME dataset over 30 repetitions under Tukey loss with $c = 4.685$.

## C. Proof of Proposition 3.3

*Proof.* Let $F_{f|S=s}(t)$ denote the conditional CDF given $S = s$. The Kolmogorov-Smirnov (KS) distance is defined as $D_{KS}(F_s, F_{s'}) = \sup_{t \in \mathbb{R}} |F_{f|S=s}(t) - F_{f|S=s'}(t)|$. The quantile function is defined as the generalized inverse of the CDF: $Q_{f|s}(\tau) = \inf\{x \in \mathbb{R} : F_{f|S=s}(x) \geq \tau\}$ for $\tau \in (0, 1)$, which is a non-decreasing and left-continuous function.

We prove the equivalence in two directions.

**1. Sufficiency ($\Longleftarrow$)**: Assume the KS distance is zero for all pairs $s, s'$. That is:

$$\sup_{t \in \mathbb{R}} |F_{f|S=s}(t) - F_{f|S=s'}(t)| = 0 \implies F_{f|S=s}(t) = F_{f|S=s'}(t), \quad \forall t \in \mathbb{R}.$$

Since the quantile function $Q(\tau)$ is uniquely determined by $F(t)$, it follows that

$$Q_{f|s}(\tau) = Q_{f|s'}(\tau), \quad \forall \tau \in (0, 1).$$

Consequently, the integrand $|Q_{f|s}(\tau) - Q_{f|s'}(\tau)|$ is identically zero. The Wasserstein distance, being the integral of this difference, is therefore zero:

$$W_1(\nu_s^f, \nu_{s'}^f) = \int_0^1 0 \, d\tau = 0.$$

Taking the supremum over all pairs yields $\mathcal{U}(f) = 0$.

**2. Necessity ( $\implies$ ):** Assume $\mathcal{U}(f) = 0$. By definition of the supremum, this implies that for any pair $s, s'$, the 1-Wasserstein distance is zero:

$$\int_0^1 |Q_{f|s}(\tau) - Q_{f|s'}(\tau)| \, d\tau = 0.$$

Since the integrand is non-negative, it must be zero almost everywhere (with respect to the Lebesgue measure on $(0,1)$). That is, there exists a set $E \subset (0,1)$ with measure $\mu_0(E) = 1$ such that $Q_{f|s}(\tau) = Q_{f|s'}(\tau)$ for all $\tau \in E$.

We extend this "almost everywhere" equality to "everywhere" equality by exploiting the properties of quantile functions. Quantile functions are, by definition, **left-continuous**. For any $\tau_0 \in (0,1)$, there exists a sequence $\{\tau_n\} \subset E$ such that $\tau_n \uparrow \tau_0$ (since $E$ is dense in $(0,1)$). By left-continuity:

$$Q_{f|s}(\tau_0) = \lim_{n \to \infty} Q_{f|s}(\tau_n) = \lim_{n \to \infty} Q_{f|s'}(\tau_n) = Q_{f|s'}(\tau_0).$$

Thus, $Q_{f|s}(\tau) = Q_{f|s'}(\tau)$ for all $\tau \in (0,1)$.

Finally, the CDF can be recovered from the quantile function via the relation $F(t) = \sup\{\tau \in (0,1) : Q(\tau) \leq t\}$. Since $Q_{f|s} \equiv Q_{f|s'}$, the sets $\{\tau : Q_{f|s}(\tau) \leq t\}$ and $\{\tau : Q_{f|s'}(\tau) \leq t\}$ are identical for any $t \in \mathbb{R}$. Therefore, their supremums are equal:

$$F_{f|S=s}(t) = F_{f|S=s'}(t), \quad \forall t \in \mathbb{R}.$$

This implies $\sup_t |F_{f|S=s}(t) - F_{f|S=s'}(t)| = 0$. $\qquad \square$

## D. Proof of Theorem 3.5

*Proof.* First, we express the conditional event for the interpolated predictor. By the definition $g_\lambda^* = \lambda f^* + (1-\lambda)g_{Q^*}^*$, we have:

$$P\left[g_\lambda^*(\boldsymbol{X}, S) \leq t \mid S = s\right]$$
$$= P\left[\lambda f^*(\boldsymbol{X}, S) + (1-\lambda)g_{Q^*}^*(\boldsymbol{X}, S) \leq t \mid S = s\right]$$
$$= P\left[\lambda F_{f^*|S=s}^{-1} \circ \left(F_{f^*|S=s}(f^*(\boldsymbol{X}, s))\right) + (1-\lambda)Q^* \circ \left(F_{f^*|S=s}(f^*(\boldsymbol{X}, s))\right) \leq t \mid S = s\right].$$

Define the random variable $U = F_{f^*|S=s}(f^*(\boldsymbol{X}, S))$. Under the condition $S = s$, $U \sim \text{Uniform}(0,1)$. Let $H_s(\tau) = \lambda F_{f^*|S=s}^{-1}(\tau) + (1-\lambda)Q^*(\tau)$, where $\tau \in [0,1]$. We then get

$$P\left[\lambda F_{f^*|S=s}^{-1}(U) + (1-\lambda)Q^*(U) \leq t \mid S = s\right] = P\left[H_s(U) \leq t \mid S = s\right].$$

Since $F_{f^*}^{-1}$ and $Q^*$ are non-decreasing and $\lambda \in [0,1]$, $H_s$ is non-decreasing and invertible. We have

$$P\left[H_s(U) \leq t \mid S = s\right] = P\left[U \leq H_s^{-1}(t) \mid S = s\right] = H_s^{-1}(t), \tag{9}$$

where the last equality follows from $U \sim \text{Uniform}(0,1)$. By Equation (9), we obtain that $H_s^{-1}(t)$ is the conditional CDF of $g_\lambda^*(\boldsymbol{X}, S)$ given $S = s$. Define $Q_s^{g_\lambda^*}$ as the conditional quantile function of $g_\lambda^*(\boldsymbol{X}, S)$ given $S = s$. By the definition of $Q_s^{g_\lambda^*}$, we have

$$Q_s^{g_\lambda^*}(\tau) = (H_s^{-1})^{-1}(\tau) = H_s(\tau) = \lambda F_{f^*|S=s}^{-1}(\tau) + (1-\lambda)Q^*(\tau).$$

Since

$$W_1(\nu_s^{g_\lambda^*}, \nu_{s'}^{g_\lambda^*}) = \int_0^1 \left|Q_s^{g_\lambda^*}(\tau) - Q_{s'}^{g_\lambda^*}(\tau)\right| d\tau,$$

we have

$$W_1(\nu_s^{g_\lambda^*}, \nu_{s'}^{g_\lambda^*}) = \int_0^1 \left|\left(\lambda F_{f^*|S=s}^{-1}(\tau) + (1-\lambda)Q^*(\tau)\right) - \left(\lambda F_{f^*|S=s'}^{-1}(\tau) + (1-\lambda)Q^*(\tau)\right)\right| d\tau$$

$$= \int_0^1 \left|\lambda \left(F_{f^*|S=s}^{-1}(\tau) - F_{f^*|S=s'}^{-1}(\tau)\right)\right| d\tau.$$

Since $\lambda \geq 0$, we can factor it out of the integral:

$$W_1(\nu_s^{g_\lambda^*}, \nu_{s'}^{g_\lambda^*}) = \lambda \int_0^1 \left| F_{f^*|S=s}^{-1}(\tau) - F_{f^*|S=s'}^{-1}(\tau) \right| d\tau$$
$$= \lambda W_1(\nu_s^{f^*}, \nu_{s'}^{f^*}).$$

Finally, taking the supremum over all pairs $s, s' \in \mathcal{S}$ yields the linear scaling of the Fairness Gap:

$$\mathcal{U}(g_\lambda^*) = \sup_{s,s'} \left( \lambda W_1(\nu_s^{f^*}, \nu_{s'}^{f^*}) \right)$$
$$= \lambda \sup_{s,s'} W_1(\nu_s^{f^*}, \nu_{s'}^{f^*})$$
$$= \lambda \mathcal{U}(f^*).$$

This completes the proof. $\qquad\square$

## E. Proof of Theorem 3.6

*Proof.* Let $J(\lambda) = \mathbb{E}[L(g_\lambda^*(\boldsymbol{X}, S), Y)]$ denote the risk associated with the interpolation parameter $\lambda$.

**1. Convexity of $J(\lambda)$.** We prove that $J(\lambda)$ is a convex function on $[0, 1]$.

Recall that

$$g_\lambda^* = (1 - \lambda)g_{Q^*}^* + \lambda f^*.$$

For any $\lambda_1, \lambda_2 \in [0, 1]$ and any $\alpha \in [0, 1]$, let $\bar{\lambda} = \alpha\lambda_1 + (1 - \alpha)\lambda_2$. Since the loss function $L(a, y)$ is convex with respect to $a$, we have

$$L(g_{\bar{\lambda}}^*, Y) = L\left(\alpha g_{\lambda_1}^* + (1 - \alpha)g_{\lambda_2}^*, Y\right)$$
$$\leq \alpha L(g_{\lambda_1}^*, Y) + (1 - \alpha)L(g_{\lambda_2}^*, Y).$$

It follows that

$$J(\bar{\lambda}) = \mathbb{E}[L(g_{\bar{\lambda}}^*, Y)]$$
$$\leq \alpha \mathbb{E}[L(g_{\lambda_1}^*, Y)] + (1 - \alpha)\mathbb{E}[L(g_{\lambda_2}^*, Y)]$$
$$= \alpha J(\lambda_1) + (1 - \alpha)J(\lambda_2).$$

Thus, the function $J(\lambda)$ is convex.

**2. Monotonicity.** By definition of $f^*$, we have $J(1) \leq J(\lambda)$ for all $\lambda \in [0, 1]$. Consider any two points $0 \leq \lambda_1 < \lambda_2 \leq 1$. We can express $\lambda_2$ as a convex combination of $\lambda_1$ and 1:

$$\lambda_2 = \theta\lambda_1 + (1 - \theta),$$

where $\theta = \frac{1-\lambda_2}{1-\lambda_1} \in [0, 1)$. By the convexity of $J(\lambda)$, we have:

$$J(\lambda_2) \leq \theta J(\lambda_1) + (1 - \theta)J(1).$$

Since $J(1) \leq J(\lambda_1)$, we have

$$J(\lambda_2) \leq \theta J(\lambda_1) + (1 - \theta)J(\lambda_1) = J(\lambda_1).$$

Thus, for any $\lambda_1 < \lambda_2$, we have $J(\lambda_2) \leq J(\lambda_1)$. $\qquad\square$

# F. Proof of Theorem 4.2

*Proof.* The proof proceeds by establishing a bijection between the set of valid quantile functions $\mathcal{Q}$ and the set of rank-preserving fair predictors $\mathcal{G}_{\text{rank}}$.

**Step 1: Sufficiency.** We prove that $g_Q \in \mathcal{G}_{\text{rank}}$.

For any quantile function $Q \in \mathcal{Q}$, we have $g_Q(\boldsymbol{x}, s) = Q(F_{f^*|s}(f^*(\boldsymbol{x}, s)))$. By Theorem 3.1, $g_Q$ satisfies demographic parity. Furthermore, since both $Q$ and $F_{f^*|s}$ are non-decreasing functions, their composition preserves the intra-group ranking of the original predictor $f^*$. Then, we get $g_Q \in \mathcal{G}_{\text{rank}}$.

**Step 2: Necessity.** We prove that for any $g \in \mathcal{G}_{\text{rank}}$, there exists a quantile function $Q \in \mathcal{Q}$ such that

$$g(\boldsymbol{x}, s) = Q\left(F_{f^*|s}\left(f^*(\boldsymbol{x}, s)\right)\right).$$

**Step 2.1: Existence of $T_s$.** Since $g$ satisfies intra-group rank-preservation with respect to $f^*$, for each group $s$, we first prove that there exist a non-decreasing transformation $T_s : \mathbb{R} \to \mathbb{R}$ such that $g(\boldsymbol{x}, s) = T_s(f^*(\boldsymbol{x}, s))$.

For a fixed group $s \in \mathcal{S}$, let $\mathcal{Y}_s = \{f^*(\boldsymbol{x}, s) \mid \boldsymbol{x} \in \mathcal{X}\}$. Define the function $T_s : \mathcal{Y}_s \to \mathbb{R}$ by the relation:

$$T_s(y) = g(\boldsymbol{x}, s) \quad \text{for any } \boldsymbol{x} \in \mathcal{X} \text{ such that } f^*(\boldsymbol{x}, s) = y. \tag{10}$$

To establish that $T_s$ is a well-defined function, we must demonstrate that the value $T_s(y)$ depends solely on $y$ and is independent of the specific choice of $\boldsymbol{x}$.

Consider an arbitrary value $y \in \mathcal{Y}_s$. Let $\mathcal{X}_y = \{\boldsymbol{x} \in \mathcal{X} \mid f^*(\boldsymbol{x}, s) = y\}$. Suppose we select two distinct representatives $\boldsymbol{x}_a, \boldsymbol{x}_b \in \mathcal{X}_y$. By definition, we have:

$$f^*(\boldsymbol{x}_a, s) = y = f^*(\boldsymbol{x}_b, s).$$

Since $f^*(\boldsymbol{x}_a, s) \geq f^*(\boldsymbol{x}_b, s)$, by the definition of intra-group rank-preservation, it implies $g(\boldsymbol{x}_a, s) \geq g(\boldsymbol{x}_b, s)$. Since $f^*(\boldsymbol{x}_b, s) \geq f^*(\boldsymbol{x}_a, s)$, it implies $g(\boldsymbol{x}_b, s) \geq g(\boldsymbol{x}_a, s)$. Combining these inequalities yields $g(\boldsymbol{x}_a, s) = g(\boldsymbol{x}_b, s)$. Thus, $g(\boldsymbol{x}, s)$ is constant on the set $\mathcal{X}_y$, meaning the assignment $T_s(y) = g(\boldsymbol{x}, s)$ yields a unique value regardless of the choice of $\boldsymbol{x}$. Hence, $T_s$ is well-defined.

**Step 2.2: Monotonicity of $T_s$.** Let $y_1, y_2 \in \mathcal{Y}_s$ such that $y_1 > y_2$. Choose any $\boldsymbol{x}_1, \boldsymbol{x}_2$ satisfying $f^*(\boldsymbol{x}_1, s) = y_1$ and $f^*(\boldsymbol{x}_2, s) = y_2$. Since $f^*(\boldsymbol{x}_1, s) > f^*(\boldsymbol{x}_2, s)$, the rank-preservation property implies:

$$g(\boldsymbol{x}_1, s) \geq g(\boldsymbol{x}_2, s).$$

By the definition of $T_s$, this is equivalent to $T_s(y_1) \geq T_s(y_2)$. Therefore, $T_s$ is monotonically non-decreasing.

**Step 2.3: Derivation of the Functional Form.** Since $g$ satisfies demographic parity, the distribution of $g(\boldsymbol{X}, S)$ is invariant across groups. Let $H$ be the CDF of $g(\boldsymbol{X}, S)$, and let $Q_H = H^{-1}$ be its corresponding quantile function.

To handle the potential non-strict monotonicity of $T_s$, we define its *generalized inverse* $T_s^{-1}$ as:

$$T_s^{-1}(t) = \sup\{y \in \mathbb{R} : T_s(y) \leq t\}.$$

Using the property of the generalized inverse for non-decreasing functions (assuming right-continuity), the event $\{T_s(Y) \leq t\}$ is equivalent to $\{Y \leq T_s^{-1}(t)\}$. We have:

$$
\begin{aligned}
H(t) &= P(T_s(f^*(\boldsymbol{X}, S)) \leq t \mid S = s) \\
&= P(f^*(\boldsymbol{X}, S) \leq T_s^{-1}(t) \mid S = s) \\
&= F_{f^*|s}\left(T_s^{-1}(t)\right).
\end{aligned}
$$

Inverting this relationship yields $t = T_s(F_{f^*|s}^{-1}(H(t)))$. By setting $y = F_{f^*|s}^{-1}(H(t))$, we get $t = T_s(y)$ and $Q_H(F_{f^*|s}(y)) = Q_H\left(F_{f^*|s}\left(F_{f^*|s}^{-1}(H(t))\right)\right) = t$. It follows that

$$T_s(y) = Q_H(F_{f^*|s}(y)).$$

By Equation (10), we have

$$g(\boldsymbol{x}, s) = Q_H(F_{f^*|s}(y)) \quad \text{for any } \boldsymbol{x} \in \mathcal{X} \text{ such that } f^*(\boldsymbol{x}, s) = y.$$

It follows that $g(\boldsymbol{x}, s) = Q_H(F_{f^*|s}(f^*(\boldsymbol{x}, s)))$ This demonstrates that any $g \in \mathcal{G}_{\text{rank}}$ must necessarily take the form $Q_H \circ F_{f^*|s} \circ f^*$.

**Conclusion.** Since every $g \in \mathcal{G}_{\text{rank}}$ corresponds to a $Q_H \in \mathcal{Q}$, minimizing the risk $\mathcal{R}$ over the function space $\mathcal{G}_{\text{rank}}$ is equivalent to minimizing $\mathcal{R}$ over the space $\mathcal{Q}$. Therefore, our solution $g_{Q^*}^*$ is the risk-minimizing predictor in the class $\mathcal{G}_{\text{rank}}$. $\square$

# G. Proof of Theorem 4.3

*Proof.* For Lipschitz continuous loss, we have the property $|L(a, y) - L(b, y)| \leq c_L |a - b|$. We bound the risk difference between the interpolated predictor and the unconstrained predictor $f^*$:

$$\begin{aligned}
\mathcal{R}(g_\lambda^*; L) - \mathcal{R}(f^*; L) &= \mathbb{E}[L(g_\lambda^*, Y)] - \mathbb{E}[L(f^*, Y)] \\
&\leq \mathbb{E}\left|L(g_\lambda^*, Y) - L(f^*, Y)\right| \\
&\leq c_L \mathbb{E}\left|g_\lambda^* - f^*\right|.
\end{aligned}$$

Substituting the expression $g_\lambda^* - f^* = (1 - \lambda)(g_{Q^*}^* - f^*)$:

$$\begin{aligned}
\mathcal{R}(g_\lambda^*; L) - \mathcal{R}(f^*; L) &\leq c_L \mathbb{E}\left|(1 - \lambda)(g_{Q^*}^* - f^*)\right| \\
&= c_L(1 - \lambda)\mathbb{E}\left|f^* - g_{Q^*}^*\right|.
\end{aligned}$$

Rearranging the terms gives the upper bound:

$$\mathcal{R}(g_\lambda^*; L) \leq \mathcal{R}(f^*; L) + c_L(1 - \lambda)\mathbb{E}\left|f^* - g_{Q^*}^*\right|.$$

This completes the proof. $\square$

# H. Proof of Theorem 4.5

*Proof.* Let

$$\mathbb{M}_n(Q) = \frac{1}{n} \sum_{i=1}^n L\left(Q\left(\widehat{F}_{\widehat{f}|S=S_i}(\widehat{f}(\boldsymbol{X}_i, S_i))\right), Y_i\right)$$

denote the empirical risk based on the estimated inputs, and let $M(Q) = \mathbb{E}_{\boldsymbol{Z}}\left[L\left(Q\left(F_{f^*|S}(f^*(\boldsymbol{X}, S))\right), Y\right)\right]$ denote the population risk based on true inputs, where $\boldsymbol{Z} = (\boldsymbol{X}, S, Y)$. Let $\Pi_n Q^*$ be the minimizer of $M(Q)$ over $Q \in \mathcal{Q}_n$. Since $\widehat{Q}$ minimizes $\mathbb{M}_n(Q)$ over $\mathcal{Q}_n$, we have $\mathbb{M}_n(\widehat{Q}) \leq \mathbb{M}_n(\Pi_n Q^*)$.

**1. Basic Inequality Decomposition** By the Margin Condition (Assumption 4.4(iv)), we have:

$$\begin{aligned}
&M(\widehat{Q}) - M(Q^*) \\
&= M(\widehat{Q}) - M(Q^*) - \left(\mathbb{M}_n(\widehat{Q}) - \mathbb{M}_n(\Pi_n Q^*)\right) + \underbrace{\mathbb{M}_n(\widehat{Q}) - \mathbb{M}_n(\Pi_n Q^*)}_{\leq 0} \\
&\leq \underbrace{\left(M(\widehat{Q}) - \mathbb{M}_n(\widehat{Q})\right) - (M(\Pi_n Q^*) - \mathbb{M}_n(\Pi_n Q^*))}_{T_1 : \text{Stochastic Process Term}} \\
&\quad + \underbrace{M(\Pi_n Q^*) - M(Q^*)}_{T_2 : \text{Approximation Error}}.
\end{aligned} \tag{11}$$

**2. Bounding the Approximation Error ($T_2$).** By Assumption 4.4(iv) and Lemma A.1 in Lu et al. (2007), we have

$$T_2 = \mathbb{E}\left|L\left(\Pi_n Q^*\left(F_{f^*|S}(f^*(\boldsymbol{X}, S))\right), Y\right) - L\left(Q^*\left(F_{f^*|S}(f^*(\boldsymbol{X}, S))\right), Y\right)\right| \leq c_2 \|\Pi_n Q^* - Q^*\|_2^2 \lesssim m_n^{-2r}.$$

**3. Bounding the Stochastic Process Term** $(T_1)$. The term $T_1$ involves the empirical process indexed by the class $\mathcal{Q}_n$. However, we must account for the nuisance parameter introduced by the first-step estimation. We decompose $T_1$ further by adding and subtracting the empirical risk evaluated at the true inputs:

$$T_1 = \underbrace{(\mathbb{P} - \mathbb{P}_n)\left[L\left(\widehat{Q}\left(F_{f^*|S}(f^*(\boldsymbol{X}, S))\right), Y\right) - L\left(\Pi_n Q^*\left(F_{f^*|S}(f^*(\boldsymbol{X}, S))\right), Y\right)\right]}_{T_{1,a}:\text{ Empirical Process}}$$

$$+ \underbrace{\mathbb{P}_n L\left(\widehat{Q}\left(F_{f^*|S}(f^*(\boldsymbol{X}, S))\right), Y\right) - \mathbb{P}_n L\left(\Pi_n Q^*\left(F_{f^*|S}(f^*(\boldsymbol{X}, S))\right), Y\right) - [\mathbb{M}_n(\widehat{Q}) - \mathbb{M}_n(\Pi_n Q^*)]}_{T_{1,b}:\text{Nuisance Error}}.$$

**Term $T_{1,a}$ (Empirical Process):** We first establish the complexity of the function class induced by the loss function. Let $\mathcal{G}_n = \{g(x, s, y) = L(Q(F_{f^*|S=s}(f^*(x, s))), y) : Q \in \mathcal{Q}_n\}$. Since the loss function $L(\cdot, y)$ is Lipschitz continuous with constant $c_2$ (Assumption 4.4(iii)), the covering number (and bracketing number) of $\mathcal{G}_n$ is controlled by that of the sieve space $\mathcal{Q}_n$. Specifically, an $\epsilon$-bracket $[Q^L, Q^U]$ for $\mathcal{Q}_n$ translates to a $c_2\epsilon$-bracket $[L(Q^L(\cdot), y), L(Q^U(\cdot), y)]$ for $\mathcal{G}_n$.

For the space of monotone I-splines of dimension $m_n$, the bracketing entropy is given by $\log N_{[]}(\epsilon, \mathcal{Q}_n, \|\cdot\|_2) \asymp m_n \log(1/\epsilon)$ in Shen & Wong (1994) (page 597). Consequently, the empirical process indexed by $\mathcal{G}_n$ satisfies the same entropy bound. Using the maximal inequality, for $\delta = \|\widehat{Q} - Q^*\|_2$:

$$\mathbb{E}\left[\sup_{\|Q-Q^*\|_2 \leq \delta}\left|(\mathbb{P} - \mathbb{P}_n)\left[L\left(Q\left(F_{f^*|S}(f^*(\boldsymbol{X}, S))\right), Y\right) - L\left(\Pi_n Q^*\left(F_{f^*|S}(f^*(\boldsymbol{X}, S))\right), Y\right)\right]\right|\right]$$

$$\lesssim \frac{1}{\sqrt{n}}\int_0^\delta \sqrt{m_n \log(1/\epsilon)}d\epsilon \asymp \sqrt{\frac{m_n}{n}}\delta.$$

**Term $T_{1,b}$ (Nuisance Error):** We expand $T_{1,b}$ explicitly:

$$|T_{1,b}| = \left|\frac{1}{n}\sum_{i=1}^n\left[L\left(\widehat{Q}\left(F_{f^*|S=S_i}(f^*(\boldsymbol{X}_i, S_i))\right), Y_i\right) - L\left(\widehat{Q}\left(\widehat{F}_{\widehat{f}|S=S_i}(\widehat{f}(\boldsymbol{X}_i, S_i))\right), Y_i\right)\right]\right.$$

$$\left. - \frac{1}{n}\sum_{i=1}^n\left[L\left(\Pi_n Q^*\left(F_{f^*|S=S_i}(f^*(\boldsymbol{X}_i, S_i))\right), Y_i\right) - L\left(\Pi_n Q^*\left(\widehat{F}_{\widehat{f}|S=S_i}(\widehat{f}(\boldsymbol{X}_i, S_i))\right), Y_i\right)\right]\right|.$$

Using the Lipschitz continuity of $L$ (Assumption 4.4(iii)) and the Triangle Inequality:

$$|T_{1,b}| \leq \frac{1}{n}\sum_{i=1}^n c_2\left|\widehat{Q}\left(F_{f^*|S=S_i}(f^*(\boldsymbol{X}_i, S_i))\right) - \widehat{Q}\left(\widehat{F}_{\widehat{f}|S=S_i}(\widehat{f}(\boldsymbol{X}_i, S_i))\right)\right|$$

$$+ \frac{1}{n}\sum_{i=1}^n c_2\left|\Pi_n Q^*\left(F_{f^*|S=S_i}(f^*(\boldsymbol{X}_i, S_i))\right) - \Pi_n Q^*\left(\widehat{F}_{\widehat{f}|S=S_i}(\widehat{f}(\boldsymbol{X}_i, S_i))\right)\right|.$$

Next, we apply the Mean Value Theorem (or Lipschitz property) to the quantile functions $Q \in \mathcal{Q}_n$. For any $Q \in \mathcal{Q}_n$, $|Q(u) - Q(v)| \leq \sup_{\tau \in [0,1]}|Q'(\tau)| \cdot |u - v|$. Thus, it yields that

$$|T_{1,b}| \leq 2c_2 \cdot \left(\sup_{Q \in \mathcal{Q}_n}\|Q'\|_\infty\right) \cdot \underbrace{\frac{1}{n}\sum_{i=1}^n\left|F_{f^*|S=S_i}(f^*(\boldsymbol{X}_i, S_i)) - \widehat{F}_{\widehat{f}|S=S_i}(\widehat{f}(\boldsymbol{X}_i, S_i))\right|}_{\Delta_{input}}.$$

We rigorously bound the input error term $\Delta_{input}$ by decomposing it into the prediction error and the distribution estimation error:

$$\Delta_{input} \leq \frac{1}{n}\sum_{i=1}^n\left|F_{f^*|S=S_i}(f^*(\boldsymbol{X}_i, S_i)) - F_{\widehat{f}|S=S_i}(\widehat{f}(\boldsymbol{X}_i, S_i))\right| \quad \text{(I)}$$

$$+ \frac{1}{n}\sum_{i=1}^n\left|F_{\widehat{f}|S=S_i}(\widehat{f}(\boldsymbol{X}_i, S_i)) - \widehat{F}_{\widehat{f}|S=S_i}(\widehat{f}(\boldsymbol{X}_i, S_i))\right|, \quad \text{(II)}$$

where $F_{\widehat{f}|S}(t)$ is defined as $P(\widehat{f}(\boldsymbol{X}, S) \leq t|S, \widehat{f})$

**Bounding Term (I):** By Markov's inequality, for any $A > 0$, we have:

$$P((\text{I}) > A\mathbb{E}[(\text{I})]) \leq \frac{1}{A}.$$

To bound $\mathbb{E}[(\text{I})]$, we introduce an independent copy $\tilde{\boldsymbol{X}}$ drawn from the same conditional distribution $\boldsymbol{X}$ given $S$. It follows that

$$F_{f^*|S}(f^*(\boldsymbol{X}, S)) = \mathbb{E}_{\tilde{\boldsymbol{X}}}\left[\mathbb{I}(f^*(\tilde{\boldsymbol{X}}, S) \leq f^*(\boldsymbol{X}, S)) \mid \boldsymbol{X}, S\right],$$

$$F_{\widehat{f}|S}(\widehat{f}(\boldsymbol{X}, S)) = \mathbb{E}_{\tilde{\boldsymbol{X}}}\left[\mathbb{I}(\widehat{f}(\tilde{\boldsymbol{X}}, S) \leq \widehat{f}(\boldsymbol{X}, S)) \mid \boldsymbol{X}, S, \widehat{f}\right].$$

Thus, the expectation of $I$ becomes

$$\mathbb{E}[(\text{I})] = \mathbb{E}\left|F_{f^*|S}(f^*(\boldsymbol{X}, S)) - F_{\widehat{f}|S}(\widehat{f}(\boldsymbol{X}, S))\right|$$

$$= \mathbb{E}\left|\mathbb{E}_{\tilde{\boldsymbol{X}}}\left[\mathbb{I}\left\{f^*(\tilde{\boldsymbol{X}}, S) \leq f^*(\boldsymbol{X}, S)\right\} - \mathbb{I}\left\{\widehat{f}(\tilde{\boldsymbol{X}}, S) \leq \widehat{f}(\boldsymbol{X}, S)\right\} \Big| \boldsymbol{X}, S, \widehat{f}\right]\right|.$$

By Jensen's inequality and the linearity of expectation, we move the absolute value inside. Let $\Delta_f(\boldsymbol{X}, S) = f^*(\boldsymbol{X}, S) - \widehat{f}(\boldsymbol{X}, S)$. We analyze the difference of indicators:

$$\mathbb{I}\left\{f^*(\tilde{\boldsymbol{X}}, S) \leq f^*(\boldsymbol{X}, S)\right\} - \mathbb{I}\left\{\widehat{f}(\tilde{\boldsymbol{X}}, S) \leq \widehat{f}(\boldsymbol{X}, S)\right\}$$

$$= \mathbb{I}\left\{f^*(\tilde{\boldsymbol{X}}, S) - f^*(\boldsymbol{X}, S) \leq 0\right\} - \mathbb{I}\left\{f^*(\tilde{\boldsymbol{X}}, S) - f^*(\boldsymbol{X}, S) \leq \Delta_f(\boldsymbol{X}, S) - \Delta_f(\tilde{\boldsymbol{X}}, S)\right\}.$$

Applying the indicator decomposition inequality $|\mathbb{I}\{a \leq 0\} - \mathbb{I}\{a \leq b\}| \leq \mathbb{I}\{|a| \leq |b|\}$ with $a = f^*(\tilde{\boldsymbol{X}}, S) - f^*(\boldsymbol{X}, S)$ and $b = \Delta_f(\boldsymbol{X}, S) - \Delta_f(\tilde{\boldsymbol{X}}, S)$, we obtain:

$$\mathbb{E}[(\text{I})] \leq \mathbb{E}\left[\mathbb{I}\left\{|f^*(\tilde{\boldsymbol{X}}, S) - f^*(\boldsymbol{X}, S)| \leq |\Delta_f(\boldsymbol{X}, S) - \Delta_f(\tilde{\boldsymbol{X}}, S)|\right\}\right]$$

$$\leq \mathbb{E}\left[\mathbb{I}\left\{|f^*(\tilde{\boldsymbol{X}}, S) - f^*(\boldsymbol{X}, S)| \leq |\Delta_f(\boldsymbol{X}, S)| + |\Delta_f(\tilde{\boldsymbol{X}}, S)|\right\}\right].$$

Using the symmetry of $\boldsymbol{X}$ and $\tilde{\boldsymbol{X}}$, the event involving the sum of errors can be bounded by twice the event involving a single error:

$$\mathbb{E}[(\text{I})] \leq 2\mathbb{E}\left[\mathbb{I}\left\{|f^*(\tilde{\boldsymbol{X}}, S) - f^*(\boldsymbol{X}, S)| \leq 2|\Delta_f(\boldsymbol{X}, S)|\right\}\right].$$

We utilize the bounded density assumption (Assumption (vi)). Let $C_{\text{pdf}}$ be the bound on the density of $f^*(\tilde{\boldsymbol{X}}, S)$.

$$\mathbb{E}[(\text{I})] \leq 2\mathbb{E}\left[P\left(|f^*(\tilde{\boldsymbol{X}}, S) - f^*(\boldsymbol{X}, S)| \leq 2|\Delta_f(\boldsymbol{X}, S)| \,\Big|\, \boldsymbol{X}, S, \widehat{f}\right)\right]$$

$$\lesssim \mathbb{E}\left[C_{\text{pdf}} \cdot 2|\Delta_f(\boldsymbol{X}, S)|\right]$$

$$\lesssim \mathbb{E}\left[|\Delta_f(\boldsymbol{X}, S)|\right] = \|\widehat{f} - f^*\|_{L_1}.$$

Given that $\|\widehat{f} - f^*\|_{L_1} \leq \|\widehat{f} - f^*\|_{L_2} = O_p(n^{-\mu})$, we have

$$\mathbb{E}[(\text{I})] = O(n^{-\mu}). \tag{12}$$

Consequently, we get $(\text{I}) = O_p(n^{-\mu})$.

**Bounding Term (II):** The second term corresponds to the supremum distance between the empirical CDF and the population CDF of the estimated scores $\widehat{f}(\boldsymbol{X}, S)$. We recall the Dvoretzky-Kiefer-Wolfowitz (DKW) inequality (Massart, 1990), which provides a tight concentration bound for empirical distribution functions.

**Theorem H.1** (Dvoretzky-Kiefer-Wolfowitz inequality). *Let $Z_1, \ldots, Z_n$ be i.i.d. real-valued random variables with cumulative distribution $F$. Let $\hat{F}$ be the empirical cumulative distribution of $Z_1, \ldots, Z_n$. Then,*

$$\mathbb{E}\|F - \hat{F}\|_\infty = \mathbb{E}\sup_{t\in\mathbb{R}}|F(t) - \hat{F}(t)| \leq \sqrt{\frac{\pi}{2n}}.$$

To simplify the proof, we assume that the scores $Z_i = \hat{f}(\boldsymbol{X}_i, S_i)$ are independent and identically distributed given $\hat{f}$, which can be satisfied by sample splitting. Applying the DKW inequality directly to Term (II), we obtain:

$$\mathbb{E}[(\text{II})] \leq \mathbb{E}\left[\sup_{t\in\mathbb{R}}\left|F_{\hat{f}|S}(t) - \widehat{F}_{\hat{f}|S}(t)\right|\right] \leq \sqrt{\frac{\pi}{2n}}. \tag{13}$$

Since $\sqrt{\frac{\pi}{2n}} = O(n^{-1/2})$, it follows by Markov's inequality that $(\text{II}) = O_p(n^{-1/2})$.

Combining the bounds for (I) and (II), we conclude:

$$\Delta_{input} = O_p(n^{-\mu}) + O_p(n^{-1/2}) = O_p(n^{-\mu}).$$

Combining the bounds, we have $\Delta_{input} = O_p(n^{-\mu})$. Regarding the derivative term, for I-splines, the derivative is controlled by the mesh size (Bernstein inequality), specifically $\sup_{Q\in\mathcal{Q}_n}\|Q'\|_\infty \lesssim m_n$. It follows that:

$$|T_{1,b}| \lesssim O_p(m_n \cdot n^{-\mu}).$$

**4. Deriving the Rate**  Substituting the bounds for $T_1$ and $T_2$ back into the inequality (11), we obtain

$$M(\widehat{Q}) - M(Q^*) \lesssim \sqrt{\frac{m_n}{n}}\delta + m_n^{-2r} + m_n \cdot n^{-\mu}.$$

Due to $\delta = \|\widehat{Q} - Q^*\|_2$ is bounded by a constant, the term $\sqrt{\frac{m_n}{n}}\delta$ is dominated by $O(\sqrt{\frac{m_n}{n}})$. Thus, the inequality simplifies to:

$$M(\widehat{Q}) - M(Q^*) \lesssim \sqrt{\frac{m_n}{n}} + m_n^{-2r} + m_n \cdot n^{-\mu}.$$

Substituting $m_n \asymp n^v$ into the bound, we get

$$\begin{aligned}M(\widehat{Q}) - M(Q^*) &= O_p\left(n^{-2vr} + n^{-\mu+v} + n^{-1/2+v/2}\right) \\ &= O_p\left(n^{-2vr} + n^{-\mu+v}\right),\end{aligned}$$

where the last inequality is due to $\mu \leq 1/2$.

**5. From Quantile Risk to Prediction Excess Risk**  Finally, we relate the convergence rate of the estimated quantile function to the excess risk of the prediction function $g$. Recall that the prediction function is defined as $g(x, s) = Q(F_{f^*|S}(f^*(x, s)))$. The risk functional $\mathcal{R}(g)$ defined in Eq. (2.3) is equivalent to the objective function $M(Q)$ when $g$ is parameterized by $Q$. Specifically:

$$\mathcal{R}(g_Q, L) = \mathbb{E}[L(g_Q(\boldsymbol{X}, S), Y)] = \mathbb{E}[L(Q(F_{f^*|S}(f^*(\boldsymbol{X}, S))), Y)] = M(Q).$$

Therefore, the excess risk of the estimator $\widehat{g}_{\widehat{Q}}(\cdot) = \widehat{Q}(\widehat{F}_{\hat{f}|S}(\hat{f}(\cdot)))$ relative to the oracle best predictor $g_{Q^*}^*$ satisfies:

$$\begin{aligned}\mathcal{R}(\widehat{g}_{\widehat{Q}}, L) - \mathcal{R}(g_{Q^*}^*, L) &= \mathcal{R}(\widehat{g}_{\widehat{Q}}, L) - M(\widehat{Q}) + M(\widehat{Q}) - M(Q^*) \\ &= \mathbb{E}[L(\widehat{Q}(\widehat{F}_{\hat{f}|S}(\hat{f}(\boldsymbol{X}, S))), Y)] - \mathbb{E}[L(\widehat{Q}(F_{f^*|S}(f^*(\boldsymbol{X}, S))), Y)] \\ &\quad + O_p\left(n^{-2vr} + n^{-\mu+v}\right).\end{aligned}$$

By the Lipschitz continuity of $L$ and $\widehat{Q}$, we obtain:

$$
\left| \mathbb{E}_{\boldsymbol{Z}}[L(\widehat{Q}(\widehat{F}_{\widehat{f}|S}(\widehat{f}(\boldsymbol{X}, S))), Y)] - \mathbb{E}_{\boldsymbol{Z}}[L(\widehat{Q}(F_{f^*|S}(f^*(\boldsymbol{X}, S))), Y)] \right|
$$
$$
\lesssim m_n \mathbb{E}_{\boldsymbol{Z}} \left| \widehat{F}_{\widehat{f}|S}(\widehat{f}(\boldsymbol{X}, S)) - F_{f^*|S}(f^*(\boldsymbol{X}, S)) \right|
$$
$$
= O_p(m_n \cdot n^{-\mu}) = O_p(n^{-\mu+v}),
$$

where the first equation follows from Equations (12) and (13), and the last equation follows from $m_n \asymp n^v$, We then conclude

$$
\mathcal{R}(\widehat{g}_{\widehat{Q}}, L) - \mathcal{R}(g_{Q^*}^*, L) = O_p\left(n^{-2vr} + n^{-\mu+v}\right).
$$

This completes the proof of Theorem 4.5.

$\square$

