# OpenReview forum: "A General Framework for Fair and Robust Regression"
_ICML.cc/2026/Conference — ICML 2026 regular_

### Official Review · Reviewer_UkfF · 2026-03-04

**Soundness:** 3
**Presentation:** 3
**Significance:** 3
**Originality:** 2
**Overall Recommendation:** 4
**Confidence:** 4

**Summary:**

This paper studies fairness in regression under demographic parity (DP) in settings with heavy-tailed noise, where standard fair regression methods based on squared error loss can be fragile. The authors propose a general post-processing framework that transforms predictions from a base regression model to satisfy DP while accommodating a broad class of robust M-estimators, including Cauchy, Huber, LAD, quantile, and Tukey losses.

The method constructs an optimal fair transformation that preserves the ranking of predictions and achieves the minimum population risk among all fair, rank-preserving predictors under the chosen loss. The paper also introduces an interpolation scheme that allows users to trade off fairness and predictive accuracy, and extends the framework to conditional demographic parity (CDP) to account for legitimate covariates correlated with sensitive attributes. The authors provide theoretical results establishing excess risk convergence rates and demonstrate through simulations and real data experiments that the proposed method improves robustness and predictive performance compared with existing fair regression approaches.

**Compliance With Llm Reviewing Policy:**

Affirmed.

**Key Questions For Authors:**

1. The paper combines several components—robust regression, fairness post-processing, and optimal quantile learning. It would be helpful if the authors could clarify which component represents the primary methodological contribution. In particular, the optimal quantile transformation appears central to the framework, but the discussion is relatively brief. Could the authors provide a clearer explanation of how this step differs from existing approaches and why it is necessary for achieving optimality under general robust loss functions? Clarification here would help assess the novelty of the method.

2. The implementation of the quantile transformation relies on an I-spline basis representation, but the paper does not reference prior literature or provide justification for this modeling choice. Could the authors provide references or additional explanation for why I-splines are appropriate in this context and whether alternative approaches were considered? A clearer connection to existing literature would improve the methodological clarity of this component.

3. The extension to conditional demographic parity appears to rely on separating predictors into those that are independent of the sensitive attribute and those that are not. Could the authors clarify how this separation is performed in practice? In many real-world datasets, a large number of predictors may be correlated with sensitive attributes, making such separation difficult. Additional discussion or empirical evidence regarding the practicality and robustness of this approach would help clarify the usefulness of the CDP extension.

4. The proposed approach relies on first estimating a robust predictor and then applying a fairness transformation. How sensitive is the final performance (both fairness and predictive risk) to the choice of the base robust estimator or loss function? Additional discussion or ablation results would help clarify how critical each component of the framework is to the overall performance.

**Limitations:**

No. The paper does not appear to include a dedicated discussion of limitations or potential societal impacts. While the work focuses on fairness in regression, it would benefit from explicitly discussing limitations of the proposed framework. For example, the method relies on a two-step post-processing procedure whose performance may depend on the quality of the initial robust predictor and on modeling choices in the quantile transformation step. The conditional demographic parity extension may also be difficult to apply in practice when many predictors are correlated with sensitive attributes. In addition, fairness defined through demographic parity may not always align with desirable outcomes in real decision-making contexts, particularly when predictive accuracy and fairness trade-offs must be considered.

**Strengths And Weaknesses:**

Strengths.
The paper addresses an important problem at the intersection of fairness and robustness in regression, particularly in settings where heavy-tailed noise makes standard squared-loss–based fair regression methods unreliable. The proposed framework is fairly general and applies to a wide class of robust M-estimators, including Cauchy, Huber, LAD, quantile, and Tukey losses, which broadens the applicability of fairness-constrained regression beyond the commonly studied squared loss setting. The authors also provide theoretical guarantees, including optimality of the proposed fair transformation among rank-preserving predictors and finite-sample convergence rates for the excess risk, which strengthens the methodological contribution. The framework further introduces a practical interpolation scheme to balance predictive accuracy and fairness and extends the approach to conditional demographic parity to account for legitimate covariates. Empirical experiments on simulated and real datasets support the proposed method and demonstrate improvements in robustness and predictive performance relative to existing fair regression approaches.

Weaknesses.
The overall methodological contribution is somewhat difficult to isolate because the framework combines several components—robust regression, fairness post-processing, and optimal quantile learning—into a two-stage procedure. While the paper suggests that the optimal quantile transformation is a key element, the novelty and role of each component are not clearly distinguished, making it unclear which aspects constitute the main methodological innovation. In particular, the discussion of the optimal quantile learning step is relatively brief despite its apparent importance in the method. The implementation relies on an I-spline basis representation, but the paper does not reference prior literature on I-splines or provide sufficient justification for this choice, which weakens the methodological grounding. In addition, the extension to conditional demographic parity (CDP) is not fully convincing. The proposed framework appears to partition predictors into variables that are independent of the sensitive attribute and those that are not, but the motivation for this separation and its practical implementation are not clearly explained. In many real applications, a large number of predictors may be correlated with sensitive attributes, making such a separation difficult or unrealistic. More discussion and justification of this design choice, as well as its practical feasibility, would strengthen the contribution.

---

> ### Author Rebuttal · Authors · 2026-03-31
>
> We sincerely thank the reviewer for the constructive feedback.
>
> **Response to Q1:**
> We clarify that our  contributions are the closed-form fair transformation, optimal quantile learning and trade-off mechanism.
>
> Previous benchmark (FRWB) relies on the fixed  transformation: $\sum\_{s'\in\mathcal{S}} p\_{s'} Q\_{f^{\ast}|s'}\left( F\_{f^\ast|s}(f^\ast(\boldsymbol{x}, s)) \right)$, which is optimal only under the squared error loss. When dealing with heavy-tailed noise under robust losses, FRWB becomes suboptimal (See Section 3.2).  To address this, we propose the fair post-processing transformation $g^\ast\_Q(\boldsymbol{x}, s) = Q \left( F\_{f^\ast|s}(f^\ast(\boldsymbol{x}, s)) \right)$ for any quantile function $Q$. This formulation significantly expands the search space for fair prediction.
>
> We then propose to estimate the optimal quantile:$$Q^\ast = \arg \min\_{Q \in \mathcal{Q}} \mathcal{R}(g^\ast\_Q; L),$$ which minimizes the risk over the entire fair class $g^\ast\_Q$. Crucially, Theorem 4.2 proves that $g^\ast\_{Q^\ast}$ is the  minimizer of the robust risk among all fair and rank-preserving function class $\mathcal{G}\_{\text{rank}}$, not just within the specific class defined by $g^\ast\_Q$. To the best of our knowledge, this is the first work to establish such global optimality guarantees for general robust losses.
>
> Furthermore, we introduce an interpolation mechanism that allows $\widehat{g}\_{\lambda}$ to satisfy a specific, user-defined unfairness level with a interpretable parameter $\lambda$ (Theorems 3.5 and 3.6).
>
> **Response to Q2:**
>
> Spline functions are popular for modeling one-dimensional functions [1]. We utilize I-splines for the following reasons:
> * **Monotonicity:** The quantile function $Q \in \mathcal{Q}$ must be monotonically increasing. We parameterize the candidate space using an I-spline basis, i.e,
> $$\\{Q | Q(\tau) = \alpha\_0 + \sum\_{j=1}^{J\_n} \alpha\_j I\_j(\tau), \alpha\_j \geq 0, j=1, \ldots, J\_n\\},$$where $I\_j(\tau)$ is the integral of a non-negative M-spline  [1] of order $d$:$$I\_j(\tau) = \int\_{0}^{\tau} M\_j(x | d, \mathbf{t}) dx.$$ Thus, the I-splines are monotonically increasing. Then, the monotonicity of $Q(\tau)$ is enforced by requiring $\alpha\_j \geq 0$ for $j = 1, \dots, J\_n$. This does not require complex derivative constraint $\frac{dQ(\tau)}{d\tau}\geq0$.
> * **Numerical Stability:** The local support property of M-splines ensures that each coefficient $\alpha\_j$ only affects the shape in the local support $[t\_j, t\_{j+d}]$ of the quantile function, preventing global oscillations and ensuring robust optimization via the constrained L-BFGS-B algorithm.
> * **Theoretical Property:** In the field of survival analysis, I-splines have been widely utilized for the estimation of cumulative hazard functions [2]. The choice of I-splines is strongly supported by non-parametric estimation theory. According to spline approximation theory, for a true quantile function $Q^{\ast}$ with smoothness order $r$, there exists a $Q \in \mathcal{Q}\_n$ such that $\sup\_{\tau}|Q^{\ast}(\tau)-Q(\tau)|=O(J\_n^{-r})$ (see Lemma A.1 in [2]). Theorem 4.5 shows, by choosing $\nu = \frac{\mu}{2r + 1}$, we establish a convergence rate of $O\_p\left(n^{-\frac{2r\mu}{2r+1}}\right)$.
>
> **Response to Q3:**
> We clarify that our framework does not need to a separation of features based on their statistical independence from the sensitive attribute. Instead, the variable $W \in \mathcal{W}$ represents legitimate explanatory factors (e.g., income level, credit rating). In practice, the selection of $W$ is guided by domain expertise and prevailing legal frameworks. While $W$ might be statistically correlated with sensitive attributes, its inclusion is legally and operationally justified. In many real-world scenarios, policy-makers accept disparities in predictions across sensitive groups, provided that such differences originate from legitimate features rather than from unjustified bias.
>
> **Response to Q4:**
>
> Our method enforces demographic parity regardless of  the chosen loss functions and  base estimators.  Tables 1 and 2 in our paper show that fairness is  insensitive to the choice of loss functions.  We conduct ablation studies under the setting of Section 6.1, where the baseline estimator $\widehat{f}$ is obtained by using deep neural networks, linear regression, random forest, and support vector machine regression.  As shown in Table 4 (see [**Anonymous Link**](https://anonymous.4open.science/r/ICML2026_anonymous_link-CAD6/README.md)),  the unfairness  is  consistently  closed to zero  across evaluated methods. Figure 4 verifies that fairness  is insensitive to  the choice of base estimator across various $\lambda$. The risk is sensitive to the  choice of base estimators  (see Table 4 and Figure 5 in the link).
>
> [1] Schumaker, L. (2007). Spline Functions: Basic Theory.
>
> [2] Lu, M., et al. (2007). Estimation of the mean function with panel count data using monotone polynomial splines.

---

> > ### Author Rebuttal · Reviewer_UkfF · 2026-04-01
> >
> > 1) The paper establishes optimality within the class of fair, rank-preserving predictors. Could you clarify how restrictive the rank-preserving assumption is, and whether optimality extends (or fails) beyond this class?
> >
> > 2) The CDP framework relies on selecting “legitimate” variables $W$ based on domain knowledge. Could you discuss how sensitive the method is to the choice of $W$, and what happens when relevant variables are omitted or misclassified.
> >
> > 3) Rank preservation is a central constraint in the framework. Could you provide more intuition on when this assumption is desirable in practice, and whether there are scenarios where relaxing it might lead to improved fairness–accuracy trade-offs.

---

> > > ### Author Response · Authors · 2026-04-03
> > >
> > > We sincerely thank the reviewer for the insightful feedback.
> > >
> > > **1. Response to Q1 & Q3:**
> > >
> > > **1.1. Expressive power and restrictiveness of $\mathcal{G}\_{\text{rank}}$.**
> > >
> > > Although the rank-preservation constraint locks the intra-group ordering of individuals, it does not restrict the shape, scale, and smoothness of predictors.  Since $$Q(F\_{f^{\ast}|s}(f^{\ast}(\boldsymbol{x}, s))) \in \mathcal{G}\_{\text{rank}}$$ holds for any quantile function $Q$,  the class $\mathcal{G}\_{\text{rank}}$ remains highly expressive.  Thus, our framework can adaptively learn  complex and non-linear transformations.
> > >
> > > Varying the choice of $Q$ allows this space to naturally contain a wide variety of fair predictors. If  $Q = \sum\_{s' \in \mathcal{S}} p\_{s'} Q\_{f^{\ast}|s'}$, the resulting predictor is the optimal fair predictor under the squared loss [1].  If $Q$ is chosen as the quantile function of an advantaged group, the transformation aligns the outcome distribution of all minority groups to match that specific advantaged group.
> > >
> > >
> > > **1.2. The desirability of rank preservation.**
> > >
> > > When applying fairness to black-box estimators, such as DNNs, a critical question naturally arises: is it justifiable to disrupt the original ordering produced by the baseline predictor $\widehat{f}$ in the pursuit of demographic parity? We argue that it is not. Furthermore, an extensive body of literature underscores the necessity of rank preservation [2,3,4], and it is highly desirable in credit scoring and hiring decisions.
> > >
> > > * **Credit Scoring:** Consider a scenario where applicant A has a higher baseline credit score given by $\widehat{f}$  than applicant B within the same minority group, due to a longer credit history. A rank-preserving fairness adjustment might shift both scores to satisfy demographic parity, but it ensures that A's final score remains higher than B's. In contrast, a fair predictor that pursues DP without rank preservation might assign a higher score to B instead of A. This creates a ``rank reversal'' that is perceived as unfair at an individual level.
> > >
> > > * **Hiring Decisions:**  In hiring decisions, the predictor $\widehat{f}$ maps professional attributes, such as years of experience and technical certifications, to a score. Consider two candidates, C1 and C2, from the same minority group applying for a technician position. If C1 has demonstrated superior technical proficiency over C2 as evaluated by the predictor $\widehat{f}$, rank preservation guarantees that C1 would not be overtaken by C2 after the fairness adjustment. This preserves the incentive for individuals to invest in skill acquisition, as their relative efforts are respected within their peer group even as the algorithm corrects systemic biases.
> > >
> > > In such contexts, fair rank preservation cautiously removes bias associated with sensitive attributes while maintaining the relative intra-group ordering of individuals, which yields logically consistent and interpretable predictions.
> > >
> > > **1.3.  Relaxing rank preservation**
> > >
> > > For strictly convex losses, such as the squared loss, the optimality in Theorem 4.2 can be extended beyond the class $\mathcal{G}\_{\text{rank}}$ [1]. In contrast, for non-convex robust losses, such optimality may no longer be guaranteed. Relaxing the rank-preservation constraint might lead to marginal reductions in risk in some special cases. However, as discussed in 1.2, deploying non-rank-preserving predictors in high-stakes scenarios introduces unjustifiable rank reversals, yielding inexplicable decisions. Therefore, maintaining rank preservation is a principled choice for robust fairness, and exploring such relaxations is an interesting direction for future research.
> > >
> > > **Response to Q2:**
> > >
> > > CDP serves as a relaxation of DP. Specifically, if the variable $W$ is omitted, the CDP constraint degenerates to the DP constraint. The DP-based method is insensitive to the choice of $W$ in terms of conditional unfairness.
> > >
> > > We conduct an experiment based on the model:
> > > $$Y = \boldsymbol{X}^\top \boldsymbol{\beta} + S + 0.8W + (2S+0.5)\epsilon,$$
> > > where $W \sim \text{Bernoulli}(0.5)$, $S \mid \boldsymbol{X}=\boldsymbol{x}, W=w \sim \text{Bernoulli}(\pi(\boldsymbol{x},w))$ and
> > > $$\pi(\boldsymbol{x},w) = \frac{1}{1 + \exp(-\boldsymbol{x}^\top \boldsymbol{\gamma}+0.5w)}.$$
> > >  Figure 6 (see link <https://anonymous.4open.science/r/ICML2026_anonymous_link-CAD6/README.md>) shows that both the standard DP-based and CDP-based methods achieve near-zero conditional unfairness.  Because the CDP-based method enforces a weaker constraint, it achieves a lower empirical risk compared to the standard DP-based method.
> > >
> > > [1]Chzhen, E., et al. (2020). Fair regression with wasserstein barycenters.
> > >
> > > [2] Plečko, D., et al. (2020). Fair data adaptation with quantile preservation.
> > >
> > > [3] Bothmann, L., et al. (2023). Causal fair machine learning via rank-preserving interventional distributions.
> > >
> > > [4] Wei, D.,  et al. (2020). Optimized Score Transformation for Fair Classification.

---

### Official Review · Reviewer_4kG6 · 2026-03-09

**Soundness:** 3
**Presentation:** 3
**Significance:** 3
**Originality:** 3
**Overall Recommendation:** 4
**Confidence:** 4

**Summary:**

This paper proposes a post-processing framework for fair and robust regression. The framework aims to address the fragility of regression models when dealing with heavy-tailed noise and their inability to simultaneously satisfy Demographic Parity (DP) constraints. The authors present a three-step scheme:(1)Robust Regression: First, a baseline predictor $\hat{f}$ is trained using robust loss functions (such as Huber, LAD, Cauchy, Tukey, etc.) to resist outliers and heavy-tailed distributions in the data.(2)Optimal Fair Transformation: Based on the Demographic Parity (DP) constraint, the baseline predictor is mapped to a fair predictor via a quantile transformation.(3)Optimal Quantile Learning: To preserve predictive accuracy while satisfying fairness, the paper utilizes monotone I-splines to approximate the quantile space and learns the optimal transformation function $\hat{Q}$ by minimizing empirical risk. The paper provides theoretical proofs for finite-sample convergence rates and validates the effectiveness of the method through simulations and the CRIME dataset.

**Compliance With Llm Reviewing Policy:**

Affirmed.

**Final Justification:**

After carefully reading the authors' rebuttal and re-evaluating the paper, my final recommendation is to maintain my positive score. Overall, this paper presents a highly modular and theoretically sound framework for fair regression. The paper's core strengths—its flexibility in adapting to various robust loss functions, the rigorous theoretical guarantees, and the highly practical $\lambda$-interpolation mechanism—make it a significant and original contribution to the fairness literature.

However, I note that the rebuttal did not extensively resolve my reservations regarding the baseline comparison (Weakness 1) and the high dependency on the baseline predictor $\hat{f}$ (Weakness 3). The comparison against FRWB on non-squared losses remains somewhat of an unequal playing field, given FRWB’s inherent design. I encourage the authors to explicitly acknowledge this architectural advantage in the final camera-ready version to ensure a fully objective empirical evaluation.

**Key Questions For Authors:**

1. **Regarding the Convexity of the Optimization Objective**: Is the optimization objective for minimizing $\mathcal{R}_n(\hat{g}_Q; L)$ in Step 3 always convex with respect to the spline coefficients $\alpha$? Specifically, for non-convex loss functions like Tukey’s biweight loss, will the L-BFGS-B algorithm fall into local optima, and how does this affect the fairness guarantee?
2. **Uncertainty Quantification and Inference of the Transformed Predictor**: The non-linear quantile transformation $\hat{g}_Q$introduced in Steps 2 and 3 may significantly alter the tail characteristics of the predictive distribution.While Step 1 mitigates the impact of heavy-tailed noise, the quantile function $\hat{Q}$ learned in Step 3 may suffer from estimation bias in extreme regions (i.e., $\tau \to 0$ or $\tau \to 1$) due to the boundary effects of monotone I-splines. Have the authors considered the robustness and stability of this method in extreme cases of long-tailed distributions?
3. **Extension to Continuous Sensitive Attributes**: The framework currently primarily handles categorical sensitive attributes $S$. In some real-world scenarios, sensitive attributes may be continuous (e.g., age). Theoretically, estimating $F_{f^*|s}$ becomes very difficult under continuous $S$; have the authors considered how to address this?

**Limitations:**

While the authors discuss the risk of DP ignoring legitimate explanatory variables and propose the Conditional Demographic Parity (CDP) extension accordingly, there is little discussion on the "robustness failure" boundaries of the model under extreme heavy-tailed distributions. Additionally, post-processing methods typically assume that the test and calibration sets are identically distributed; the ability to maintain fairness under potential distribution shifts has not been explored.

**Strengths And Weaknesses:**

**Strengths:**

1. **Generality & Modularity**: Compared to existing fair regression methods (such as FRWB, which is limited to squared error loss), this framework demonstrates excellent flexibility and can seamlessly adapt to various robust loss functions, including Cauchy and Tukey loss.
2. **Theoretical Soundness**: The paper not only proves optimality within the class of rank-preserving fair predictors (Theorem 4.2) but also establishes explicit convergence rates for estimators parameterized by Deep Neural Networks (DNN) (Theorem 4.5).
3. **Practical Utility**: The introduced interpolation mechanism (Theorems 3.5 & 3.6) has significant practical value, allowing policymakers to linearly adjust the degree of unfairness via $\lambda$ to find a balance between legal constraints and predictive utility.
4. **Presentation**: The paper is rigorously structured; Figure 2 clearly illustrates the methodological workflow, and the trade-off curves in the experimental section are highly persuasive.

**Weaknesses:**

1. **Potential Bias in Comparison with the Wasserstein Barycenter Baseline**:

   The most critical weakness is the substantial overlap with **Distributional Conformal Prediction (DCP)** (Chernozhukov et al., 2021).

   - In numerical experiments, the authors use FRWB (Chzhen et al., 2020) as the primary baseline and claim their method outperforms FRWB across all robust losses.
   - However, FRWB was originally designed as an optimal solution for squared error loss. Although the authors attempted to adapt a robust regression starting point for FRWB in Step 1, the core of the FRWB algorithm (Wasserstein Barycenter) was not designed to optimize non-squared losses.
   - This comparison may constitute a "straw man" argument against FRWB, as the proposed method explicitly optimizes for specific robust losses via $Q$-learning in Step 3, a capability the baseline lacks.

2. **Potential Computational Complexity Constraints**:

   - Step 3 involves minimizing empirical risk over the spline space $\mathcal{Q}_n$ with non-negativity constraints. As the number of strata for the control variable $W$ increases (for the CDP extension), the optimization problem must be solved independently for each stratum.
   - The paper does not discuss the competitiveness of this framework's training costs compared to standard end-to-end training methods when the data scale is massive or sensitive group partitions are extremely fine.

3. **High Dependency on the Baseline Predictor $\hat{f}$**:

   - As a post-processing framework, the final performance depends heavily on the quality of $\hat{f}$ from Step 1.
   - If the predictions of $\hat{f}$ are inherently unstable for certain protected groups, the post-processing transformation might amplify these errors; the sensitivity analysis regarding the pre-training quality of $\hat{f}$ and final fairness stability is insufficient.

---

> ### Author Rebuttal · Authors · 2026-03-31
>
> We appreciate the reviewer's  constructive suggestions.
>
> **Response to W1:**
> We clarify that our work is distinct from Distributional Conformal Prediction (DCP).
> * **Research Objectives**: We aim to obtain a fair prediction under general robust loss functions. DCP focuses on constructing a interval $\widehat{\mathcal{C}}\_{(1-\alpha)}$ that satisfies $P(Y\_{n+1} \in \widehat{\mathcal{C}}\_{(1-\alpha)}(\boldsymbol{X\_{n+1}}) \mid \boldsymbol{X\_{n+1}}=\boldsymbol{x}) \geq 1-\alpha+o\_P(1)$. In fact, the boundaries of the interval $\widehat{\mathcal{C}}\_{(1-\alpha)}(\boldsymbol{X\_{n+1}})$ do not satisfy demographic parity.
> * **Methodologies**: We propose a novel transformation $Q^{\ast} ( F\_{f^{\ast}|s}(f^{\ast}(\boldsymbol{x}, s)) )$ to enforce fairness, where
> $$Q^{\ast} = \arg \min\_{Q \in \mathcal{Q}} \mathcal{R}(g^{\ast}\_{Q}; L).$$
> To the best of our knowledge, this is the first work that uses such a form to obtain a fair pediction. We also provide a trade-off estimator $g\_\lambda^{\ast}(\boldsymbol{x}, s)$ that satisfies the specific unfairness level $\frac{\mathcal{U}(g\_\lambda^{\ast})}{\mathcal{U}(f^{\ast})}=\lambda$. Consequently, the interpolation parameter $\lambda$ has a clear and interpretable meaning. In contrast, DCP leverages $F(y| \boldsymbol{x})=P(Y\leq y| \boldsymbol{X}=\boldsymbol{x})$ to define conformity scores $V\_i=|F(y\_i|\boldsymbol{x}\_i)-1/2|$.
> * **Theoretical Contributions**: We prove that $g^\ast\_{Q^\ast}$ is the minimizer of the robust risk among all fair and rank-preserving predictors. This is the first work to establish such optimality guarantees  (Theorem 4.2). Furthermore, we establish a  convergence rate for the excess risk while DCP prove the conditional validity of $\widehat{\mathcal{C}}\_{(1-\alpha)}$.
>
> **Response to W2:**
> The estimation of $\widehat{Q}$ is highly efficient. Table 2 and 5 (see [**Anonymous Link**](https://anonymous.4open.science/r/ICML2026_anonymous_link-CAD6/README.md)) shows the runtime of estimating $\widehat{g}\_{\widehat{Q}}$ under the settings of Section 6.1.
>
> **Response to W3:**
> We conducted ablation studies under the setting of Section 6.1, where the baseline estimator $\widehat{f}$ is obtained by using deep neural networks, linear regression, random forest and support vector machine regression  (see Table 4, Figure 4 and Figure 5 ).
>
> **Response to Q1:**
> If $L$ is convex, the optimization objective $\mathcal{R}(g^{\ast}\_{Q}; L)$ is convex with respect to $Q$ (i.e., $\alpha$). When utilizing the non-convex Tukey’s biweight loss, the L-BFGS-B algorithm may converge to a local optimum. However, this does not affect the fairness guarantee. This is because Theorem 3.1 always holds even if the estimator $\widehat{Q}$ deviates significantly from $Q^{\ast}$, which has been validated in Tables 1 and 2 of our paper.
>
> **Response to Q2:** To mitigate numerical instability near the boundaries, we utilize the natural I-splines method [1], which confines $Q$ at the boundary points to be linear and hence reduces the oscillation at the boundary. Specifically, we incorporate a penalty into the  objective, i.e.,
> $$\widehat{Q} = \arg \min \_{Q \in \mathcal{Q}\_n} \mathcal{R}\_n(\widehat{g}\_Q; L)+\gamma[ Q^{\prime \prime}(0)^2+ Q^{\prime \prime}(1)^2],$$
> where $\gamma> 0$. The penalty forces $\widehat{Q}$ to be linear at the boundary $0$ and $1$. We conducte an experiment under the setting of Section 6.1 and  replace the distribution of noise with a Student's $t$-distribution with $1.5$ degrees of freedom. Figure 3 shows that the estimator $\widehat{Q}$ obtained by the natural I-spline is much smoother than the standard I-spline at the boundaries. Table 3 shows that the natural I-spline achieves a lower unfairness and standard deviation than the standard I-Spline.
>
>
> **Response to Q3:** The proposed  method can be extended to  continuous sensitive attributes. For estimating $F_{f^{\ast}|s}$,
>  we  consider a kernel-based estimator [2]:
> $$\widetilde{F}\_{\hat{f}|s}(t) = \frac{\sum\_{i=1}^n W\_{h}\left({t-\hat{f}(\boldsymbol{X}\_i,S\_i)}\right) L\_{b}(s,S\_i)}{\sum\_{i=1}^n L\_{b}(s,S\_i)},$$
> where $W\_{h}\left(u\right)=\int\_{-\infty}^{u/h} K\left({z}\right) dz$, $K\left(z\right)$ is a Gaussian kernel, $L\_{b}(s, t)$ is  a standard kernel function (that is, a symmetric, univariate probability density), such as radial basis function kernel. Then, following the procedure in our paper, we can obtain a fair prediction.
>
>
> [1]Schumaker, L. Spline functions: basic theory.
>
> [2] Hall P, et al. Cross-validation and the estimation of conditional probability densities.

---

> > ### Author Rebuttal · Reviewer_4kG6 · 2026-04-02
> >
> > I am grateful for the reviewer's time and the detailed feedback. The additional information has been very helpful in solidifying my understanding and final assessment of the paper.

---

> > > ### Author Response · Authors · 2026-04-03
> > >
> > > Dear Reviewer,
> > >
> > > We would like to express our sincere gratitude for your comments and the time you have dedicated to evaluating our work. We are encouraged to hear that our previous responses helped solidify your understanding of the paper.
> > >
> > > Regarding the remaining points that were noted as "partially resolved," we would appreciate the opportunity to address them in greater detail. To ensure we address your concerns comprehensively, could you kindly point out the specific areas or follow-up questions you have in mind?
> > >
> > > We are more than happy to provide any further information or conduct additional experiments to resolve your concerns.
> > >
> > > Best regards,
> > >
> > > The Authors

---

### Official Review · Reviewer_WBE8 · 2026-03-10

**Soundness:** 3
**Presentation:** 3
**Significance:** 3
**Originality:** 3
**Overall Recommendation:** 3
**Confidence:** 3

**Summary:**

This paper proposes a general *post-processing* framework for fair and robust regression. The method operates in three steps: 1) Train a base predictor $\\widehat{f}$ using any robust loss (e.g., Cauchy, Huber, LAD, Quantile, Tukey). 2) Transform $\\widehat{f}$ via the group-specific empirical conditional CDF $\\widehat{F}\_{\\widehat{f}|s}$ to obtain a candidate predictor satisfying demographic parity (DP). 3) Crucially, instead of using a fixed target distribution (e.g., the Wasserstein barycenter as in prior work), the framework *learns* an optimal quantile function $\\widehat{Q}$ (parameterized via monotone I-splines) that minimizes the empirical robust risk, yielding the final fair predictor $\\widehat{g}\_{\\widehat{Q}}$. The core theoretical contributions are: proving that the learned predictor is *optimal* (minimizes risk) among all rank-preserving fair predictors (Theorem 4.2); establishing non-asymptotic convergence rates for the excess risk (Theorem 4.5); and designing an interpolation mechanism $g^ * \_\\lambda = \\lambda f^ *  + (1-\\lambda)g^ * \_{Q^ * }$ that provides a precise, theoretically-guaranteed trade-off where unfairness grows linearly and risk decreases monotonically with $\\lambda$ (Theorems 3.5, 3.6). The framework is extended to conditional demographic parity (CDP). Experiments on synthetic data and the Communities and Crime (CRIME) dataset show the method achieves significantly lower unfairness than the strong fair baseline (FRWB) while maintaining *lower or comparable* predictive risk across five robust loss functions.

**Compliance With Llm Reviewing Policy:**

Affirmed.

**Key Questions For Authors:**

1. The three-step estimation procedure requires fitting the empirical conditional CDFs $\widehat{F}_{\widehat{f}|s}$ for each group. In practice, with a large number of sensitive attribute groups $|\mathcal{S}|$ or small group sample sizes, this estimation could become noisy. Could you comment on the robustness of the method in such scenarios? Are there practical strategies (e.g., smoothing techniques, sharing information across groups) to mitigate potential issues, and do the theoretical guarantees accommodate such settings?

2. The optimization for the optimal quantile function $\widehat{Q}$ involves a constrained convex problem over I-spline coefficients. Could you provide more details on the computational cost of this step in practice? How does the runtime scale with the number of spline basis functions $J_n$, sample size $n$, and number of groups $|\mathcal{S}|$? A brief discussion on computational scalability would help practitioners.

3. Theorem 3.1 assumes the conditional CDFs $F_{f^{*}|S=s}$ are continuous. In practice, with finite data and/or when the base predictor $\widehat{f}$ outputs discrete values, the empirical CDF $\widehat{F}_{\widehat{f}|s}$ is a step function. How is this handled in your implementation (e.g., via smoothing or tie-breaking)? Did you observe any practical performance issues stemming from this discontinuity?

4. The interpolation parameter $\lambda$ provides a smooth trade-off. In a real-world deployment, how would you recommend a policy-maker or practitioner to choose $\lambda$? Could your framework be extended to facilitate a data-informed or utility-driven selection of $\lambda$, perhaps by incorporating a budget on the allowable unfairness?

**Limitations:**

The paper clearly states its assumptions and scope. The following points could be discussed to provide a more complete view of the work's boundaries:
1. **Estimation of Conditional Distributions:** The framework's performance hinges on accurately estimating the conditional CDFs $F\_{f^{*}|s}$ (or their empirical counterparts). The theory (Assumption 4.4(vi)(vii)) and the DKW inequality bound this error, but in practice, for groups with very small sample sizes, this remains a challenge. The extension to CDP further requires estimating conditional-on-$W$ distributions, which exacerbates the data scarcity problem for fine-grained strata.
2. **Rank-Preservation Assumption:** The optimality guarantee (Theorem 4.2) is within the class of rank-preserving fair predictors. While this is a reasonable and often desirable constraint (preserving intra-group rankings), there might be scenarios where achieving fairness requires altering the relative order of individuals within a group. The framework does not cover such cases.
3. **Choice of Spline Complexity:** The theoretical rate depends on the smoothness of $Q^ * $ and the spline knot number $m_n$. In practice, choosing $m_n$ (or equivalently, $J_n$) is a model selection problem. A discussion on the practical sensitivity of results to this choice and potential cross-validation strategies would be helpful.
4.  **Societal Impact Consideration:** While the method aims to promote fairness, the authors should briefly acknowledge that technical fairness solutions are only one component of responsible deployment. The choice of sensitive attribute, the definition of the legitimate covariate $W$ for CDP, and the broader societal context are critical. Over-reliance on demographic parity might sometimes be inappropriate, and the method's output should be interpreted with domain expertise.

**Strengths And Weaknesses:**

1. **Soundness:** The paper is technically sound and rigorous. The theoretical analysis is comprehensive: Theorem 3.1 correctly generalizes the fair transformation, the optimality proof (Theorem 4.2) is elegant, and the convergence analysis (Theorem 4.5) carefully handles errors from both the base predictor and spline approximation. The experimental methodology is appropriate. Simulations validate theoretical properties under controlled settings, and the CRIME dataset effectively motivates the real-world problem of heavy-tailed errors combined with fairness concerns. Results are reported with standard deviations over multiple runs. The authors fairly demonstrate that their method achieves drastically lower unfairness than the fair baseline (FRWB) while maintaining comparable or even lower risk, which aligns with the expected fairness-accuracy trade-off.

2. **Presentation:** The paper is clearly written and well-structured. The logical flow from motivation to methodology, theory, and experiments is excellent. However, the presentation is dense with intertwined concepts (optimal quantile learning, rank-preservation, I-splines, $\lambda$-interpolation). A high-level schematic figure illustrating the three-step estimation pipeline and the trade-off mechanism would greatly improve accessibility for a broader audience. The related work section adequately covers the necessary literature.

3. **Significance:** The paper addresses a highly relevant and practical problem: building regression models that are both fair and robust to heavy-tailed noise, which is common in critical applications like criminal justice. The proposed framework is *general* (applicable to a wide class of robust losses) and *practical* (a post-processing method). The key advancement is providing, for the first time, a unified framework with optimality guarantees and convergence rates for this general setting, moving beyond prior work limited to specific losses (e.g., squared, quantile). The explicit, theoretically-characterized fairness-risk trade-off mechanism is a valuable feature for practitioners. This work represents a substantial contribution to the fields of fair and robust machine learning.

4. **Originality:** The originality of the paper is high. The most original aspects are: 1) Identifying and formally solving the *optimal quantile learning* problem for general robust losses under DP, which subsumes the fixed Wasserstein barycenter solution as a special case; 2) Providing the first optimality guarantee (Theorem 4.2) and convergence rates (Theorem 4.5) for this general problem; and 3) Designing the interpolated predictor with provable linear-monotonic trade-off properties. The work is a non-trivial and principled generalization of the post-processing fair regression paradigm, synthesizing ideas from optimal transport, robust statistics, and nonparametric estimation into a cohesive and theoretically-grounded framework.

---

> ### Author Rebuttal · Authors · 2026-03-27
>
> We thank the reviewer for the insightful comments and constructive feedback.
>
> **Response to Q1 & Q3:**
>
> To solve the challenge posed by a large number of sensitive attribute groups and the discontinuity of $\widehat{F}\_{\widehat{f}|s}$, we consider the kernel-based  estimator [1],
> $$\widetilde{F}\_{\hat{f}|s}(t) = \frac{\sum\_{i=1}^n W\_{h}\left({t-\hat{f}(\boldsymbol{X}\_i,S\_i)}\right) L\_{b}(s,t)}{\sum\_{i=1}^n L\_{b}(s,S\_i)},$$
>
> where $W\_{h}\left(u\right)=\int\_{-\infty}^{u/h} K\left({z}\right) dz$, $K\left(z\right)$ is a Gaussian kernel, and $L\_{b}(s, t)$ is the Wang-van Ryzin kernel, i.e.,
> $$L\_{b}(s, t) = \begin{cases} 1 - b& \text{if } t = s \\\\ \frac{1-b}{2}b^{|t-s|} & \text{if } t \neq s,\end{cases}$$
> The new estimator $\widetilde{F}\_{\hat{f}|s}(t)$ is continuous with respect to $t$ and utilizes all the data.
>
> We conducted an experiment based on the model: $$Y=\boldsymbol{\beta}\boldsymbol{X}+0.1(S-35) +\epsilon$$ with $P(S=s)=1/|\mathcal{S}|$, where $s\in\{25,26,\ldots, 45\}$. All other settings remain consistent with Section 6.1.  Table 1 and Figure 1 (see link [**Anonymous Link**](https://anonymous.4open.science/r/ICML2026_anonymous_link-CAD6/README.md)) shows that the kernel-based estimator performs similarly to $\widehat{F}\_{\hat{f}|s}(t)$ in terms of mean risk and unfairness, while yielding a slightly lower standard deviation.
>
> Theoretically, the convergence rates of excess risk can be established by considering the decomposition
> $$\begin{align\*} &|\widetilde{F}\_{\widehat{f}|s}(\widehat{f}(\boldsymbol{X}, s))-F\_{f^{\ast}|s}(f^{\ast}(\boldsymbol{X}, s))| \\\\ \leq&  |\widetilde{F}\_{\widehat{f}|s}(\widehat{f}(\boldsymbol{X}, s))-F\_{\widehat{f}|s}(\widehat{f}(\boldsymbol{X}, s))|+|F\_{\widehat{f}|s}(\widehat{f}(\boldsymbol{X}, s))-F\_{f^\ast|s}(f^\ast(\boldsymbol{X}, s))| \\\\ =&I+II. \end{align\*}$$
>
> Term I can be controlled by the square root of mean integrated squared error derived in [1]. Term II can be controlled by $|\widehat{f}(\boldsymbol{X}, s)-{f}^\ast(\boldsymbol{X}, s)|$, which is shown in the proof of Theorem 4.5.
>
>
> **Response to Q2:**
>
> The estimation of $\widehat{Q}$ is highly efficient.  Table 2 and 5 show the runtime of estimating $\widehat{g}\_{\widehat{Q}}$ with various $n, J\_n$ and $|\mathcal{S}|$ under the settings of Section 6.1.
>
> **Response to Q4:**
>
> First, the ratio $\lambda = \frac{\mathcal{U}(g\_\lambda^{\ast})}{\mathcal{U}(f^{\ast})}$ represents the relative reduction in unfairness compared to $f^{\ast}$, which is interpretable.
>
> Second, by the definition of $\mathcal{U}(g\_\lambda^{\ast})$, we get:
> $$\begin{align\*} \sup\_{s, s' \in \mathcal{S}} \left| \mathbb{E}[g\_\lambda^{\ast}(\boldsymbol{X},S) \mid S=s] - \mathbb{E}[g\_\lambda^{\ast}(\boldsymbol{X},S) \mid S=s'] \right| \leq \sup\_{s, s' \in \mathcal{S}} \int\_{0}^{1} \left| Q\_{g\_\lambda^{\ast}|s}(\tau) - Q\_{g\_\lambda^{\ast}|s^{\prime}}(\tau) \right| d\tau =\mathcal{U}(g\_\lambda^{\ast}). \end{align\*}$$
> Suppose $g\_\lambda^{\ast}(\boldsymbol{x},s)$ represents the approved loan amount. If a policy-maker requires the average disparity between any two demographic groups to be less than 500, they can directly enforce it by setting the constraint $\mathcal{U}(g\_\lambda^{\ast}) \leq 500$. It leads to the optimal solution $\lambda^{\ast} = \min\left(1, \frac{500}{\mathcal{U}(f^{\ast})}\right)$.
>
> Finally, assume there exists a budget function $\mathcal{B}(g\_\lambda^{\ast})$ determined by policymakers, such as $\mathcal{B}(g\_\lambda^{\ast}) = \mathcal{R}(g\_\lambda^{\ast}; L)$. We can identify the optimal $\lambda$ that balancing the pair $[\mathcal{U}(g\_\lambda^{\ast}), \mathcal{B}(g\_\lambda^{\ast})]$ by the concept of knee point [2].
> Let $u = \mathcal{U}(g\_\lambda^{\ast})$ and then get $\lambda(u) = \frac{u}{\mathcal{U}(f^{\ast})}$. This yields the utility-budget pair $[u, \mathcal{B}(g\_{\lambda(u)}^{\ast})]$. Let $\mathcal{T}(u) = \mathcal{B}(g\_{\lambda(u)}^{\ast})$, which represents the budget at unfairness level $u$.
> The knee point of $\mathcal{T}(u)$ maximizes its curvature, i.e,
> $$u\_0=\arg \max \frac{\mathcal{T}^{\prime\prime}(u)}{[1+\mathcal{T}^{\prime}(u)^2]^{1.5}}.$$
> Mathematically, the knee point represents the farthest point of $\mathcal{T}(u)$ away from a straight line $y=1-x$. As $\lambda$ decreases in $[0, \lambda(u\_0)]$, the budget $\mathcal{B}(g\_{\lambda}^{\ast})$ increases at a faster rate than the reduction in $\mathcal{U}(g\_{\lambda}^{\ast}).$
> We applied the method to the CRIME dataset and get f $\lambda(u\_0)=0.52$, which is shown in Figure 2.
>
>
> [1] Hall P, Racine J, Li Q.  (2004) Nonparametric estimation of conditional CDF and quantile functions with mixed categorical and continuous data.  JASA, 99: 1015-1026.
>
> [2] Satopaa, V., Albrecht, J., Irwin, D.,  Raghavan, B. (2011) Finding a" kneedle" in a haystack: Detecting knee points in system behavior.  The 31st international conference on distributed computing systems workshops IEEE,  166-171.

---

### Decision · Program_Chairs · 2026-04-30

**Decision:**

Accept (regular)

**Comment:**

This paper studies fair regression and proposes a general post-processing framework that applies to a wide class of robust M-estimators. The work is well-motivated and addresses an important gap in the literature, as most existing fair regression methods rely on squared loss and are therefore not robust to heavy-tailed noise. The paper is technically strong, providing rigorous theoretical guarantees, including optimality within the class of rank-preserving predictors and convergence rates. Moreover, the proposed interpolation mechanism offers a practical and interpretable way to balance fairness and predictive accuracy. The overall framework is general and modular, which enhances its potential impact.

A key issue, however, is that the source of novelty is somewhat difficult to isolate, as the framework combines multiple components, from robust regression, fairness post-processing, to optimal quantile learning. More clearly delineating the individual contributions of these components would be helpful. In addition, there are concerns regarding the fairness of the empirical comparisons: the primary baseline (FRWB) is inherently designed for squared loss, making comparisons under alternative robust losses potentially less informative.

Overall, despite these limitations, the paper presents a meaningful and technically solid contribution to fair and robust machine learning. The strengths outweigh the weaknesses, and I recommend a weak accept, with the expectation that the authors will clarify the sources of novelty and strengthen the empirical positioning in the final version.